# Pomeranchuk instability from electronic correlations in CsTi₃Bi₅ kagome metal

Chiara Bigi [1,17] ✉, Matteo Dürrnagel [2,3,17] ✉, Lennart Klebl[2,4], Armando Consiglio[5], Ganesh Pokharel [6,7], Marta Zonno [1], François Bertran [1], Patrick Le Fèvre [8], Thomas Jaouen [8], Hulerich C. Tchouekem[8], Pascal Turban [8], Alessandro De Vita [9], Jill A. Miwa[10], Justin W. Wells[11], Dongjin Oh [12], Riccardo Comin[12], Ronny Thomale [2], Ilija Zeljkovic [13], Brenden R. Ortiz [14], Stephen D. Wilson [6], Giorgio Sangiovanni [2], Federico Mazzola [15] ✉ & Domenico Di Sante [5,16] ✉

Electronic nematicity, the spontaneous breaking of rotational symmetry, has emerged as a key instability in correlated quantum systems. CsTi₃Bi₅, a kagome metal of the AV₃Sb₅ (A = K, Rb, Cs) family, hosts rich unconventional electronic phases, yet the origin of its nematicity remains unsettled. Here, we combine polarization-dependent angle-resolved photoemission spectroscopy with functional renormalization group calculations on a fully interacting ab initio model. We reveal an orbital-selective nematic deformation in the low-energy band structure and identify a finite angular momentum (d-wave) Pomeranchuk instability driven by electronic correlations in specific orbital channels and detuning from Van Hove singularities. Our results establish a direct link between orbital selectivity and symmetry-breaking instabilities in CsTi₃Bi₅, providing a microscopic framework for nematic order in kagome systems.

Kagome metals represent a remarkable class of materials that has garnered significant attention in condensed matter physics due to the richness of observed correlated phases. Owing to the unique topological properties of the underlying kagome lattice, highly itinerant electrons naturally display a variety of massless Dirac-like states, Van Hove singularities (VHss) with partial sublattice polarisation[1–3], and compact localised states with dispersionless bands within a single band structure[4–7]. Combined with the inherent geometrical frustration of any interacting model on the kagome lattice, this provides ideal conditions for the realisation of many exotic phases long sought in other correlated electron platforms including persistent loop current formation[8–11] and superconducting pairing modulations[12–15]. While the experimental evidence for these phases is still subject of ongoing debates, electronic nematicity, i.e., the breaking of rotational

¹Synchrotron SOLEIL, L'Orme des Merisiers, Saint-Aubin, France. ²Institute for Theoretical Physics and Astrophysics, University of Würzburg, Würzburg, Germany. ³Institute for Theoretical Physics, ETH Zürich, Zürich, Switzerland. ⁴Institute for Theoretical Physics, Universität Hamburg, Hamburg, Germany. ⁵CNR - Istituto Officina dei Materiali (IOM), Trieste, Italy. ⁶Materials Department, University of California Santa Barbara, Santa Barbara, CA, USA. ⁷Perry College of Mathematics, Computing, and Sciences, University of West Georgia, Carrollton, GA, USA. ⁸Univ Rennes, IPR Institut de Physique de Rennes, Rennes, France. ⁹Institut für Physik und Astronomie, Technische Universität Berlin, Berlin, Germany. ¹⁰Department of Physics and Astronomy, Interdisciplinary Nanoscience Center, Aarhus University, Aarhus C, Denmark. ¹¹Department of Physics and Centre for Materials Science and Nanotechnology, University of Oslo (UiO), Oslo, Norway. ¹²Department of Physics, Massachusetts Institute of Technology, Cambridge, MA, USA. ¹³Department of Physics, Boston College, Chestnut Hill, MA, USA. ¹⁴Materials Science and Technology Division, Oak Ridge National Laboratory, Oak Ridge, TN, USA. ¹⁵Department of Physics and Astronomy 'Galileo Galilei', University of Padova, Padova, Italy. ¹⁶Department of Physics and Astronomy, University of Bologna, Bologna, Italy. ¹⁷These authors contributed equally: Chiara Bigi, Matteo Dürrnagel. ✉e-mail: chiara.bigi@synchrotron-soleil.fr; matteo.duerrnagel@uni-wuerzburg.de; federico.mazzola@unipd.it; domenico.disante@unibo.it

symmetry in the charge ordered state, strikes out as one of the few universal features across all kagome compounds[16–23]. Despite its critical relevance also for the subsequent superconducting transition at low temperature[15,24–26], the origin and microscopic mechanism governing nematicity has remained elusive, necessitating focused investigation.

Among the diverse array of kagome systems, the first synthesized $AV_3Sb_5$ family (where A = K, Rb, Cs)[27,28] has remained the most extensively studied. However, the simultaneous emergence and intertwining of the various correlated electronic phases induced by a lack of scale separation between different symmetry breaking phases poses a significant challenge to the understanding of the underlying mechanisms driving the various many-body effects and remains a current thread to any thorough assessment of kagome metals[29,30].

In this study, we utilize $CsTi_3Bi_5$ as a unique platform to explore the phenomenon of rotational symmetry breaking on the kagome lattice. The absence of an accompanying translational symmetry breaking typical of other '135' kagome metals renders this compound an ideal experimental setting for a detailed study of nematicity[20,31,32]. To elucidate the underlying mechanisms, we adopt a synergistic approach that combines light-polarization-dependent angle-resolved photoelectron spectroscopy (ARPES) with ab initio based field theoretical methods. Our findings reveal that electronic nematicity in $CsTi_3Bi_5$ reduces the symmetry of the system through an orbital-selective mechanism, with dominant contributions from the planar $d_{xy}$ and $d_{x^2-y^2}$ orbitals of Ti. Crucially, our results indicate a purely electronic origin of this effect, stemming from the frustrated long-range Coulomb repulsion and considerable VHs detuning from the Fermi level. This supports the existence of a $d$-wave Pomeranchuk instability (PI) in $CsTi_3Bi_5$, characterized by the spontaneous breaking of point group symmetry driven by the divergence of an associated susceptibility of the electronic system[33], and a sublattice charge imbalance. In this respect, our findings notably differ from the prevailing view that nematic charge order in titanium-based kagome metals arises through

a bond-type order with zero total momentum[20,31]. That interpretation originated from observations of the momentum-dependent character of nematic Fermi surface distortions, which were incorrectly claimed to be incompatible with a straightforward charge imbalance across unit cell sites[20], as well as from many-body calculations that favored non-local charge orders[34].

Recently, PIs have sparked recurrent interest. In the context of multi-layer graphene[35,36], the PI is realized as a partial valley polarisation and results in exotic half and quarter-metal states where the isospin provides the additional ingredient to realize the spontaneous rotational symmetry breaking[37]. The surface states of topological elemental arsenic ($\alpha$-As) have now also been reported to support a genuine PI[38], despite weak correlations. Our work on $CsTi_3Bi_5$, on the other hand, puts forward PI as a generic instability in hexagonal lattice systems with a non-trivial sublattice degree of freedom, advancing our understanding about the microscopic mechanisms governing electronic nematicity.

## Results

$CsTi_3Bi_5$ exemplifies the defining structure of the '135' kagome metals, being isostructural with the extensively studied V-based compounds $AV_3Sb_5$. Within the Bi-Ti plane, the kagome lattice formed by Ti atoms (Fig. 1a) is pivotal in establishing a complex electronic structure characterized by itinerant Dirac-like states, VHss, and flat bands[34,39–41]. Our angle-resolved photoemission spectroscopy (ARPES) measurements unveil sharply defined bands that disperse across the Fermi level, underpinning the metallic nature of the band structure and resulting in a multifaceted Fermi surface (Fig. 1c) defined by multiple sheets with an overall hexagonal geometry. Importantly, the experimental spectra illustrated in Fig. 1d,e exhibit concordance with predictions from first-principles calculations, that is even enhanced when taking into account surface termination effects (Supplementary Note I), thereby reinforcing the reliability of our findings. We emphasize that these data are obtained through the integration of ARPES

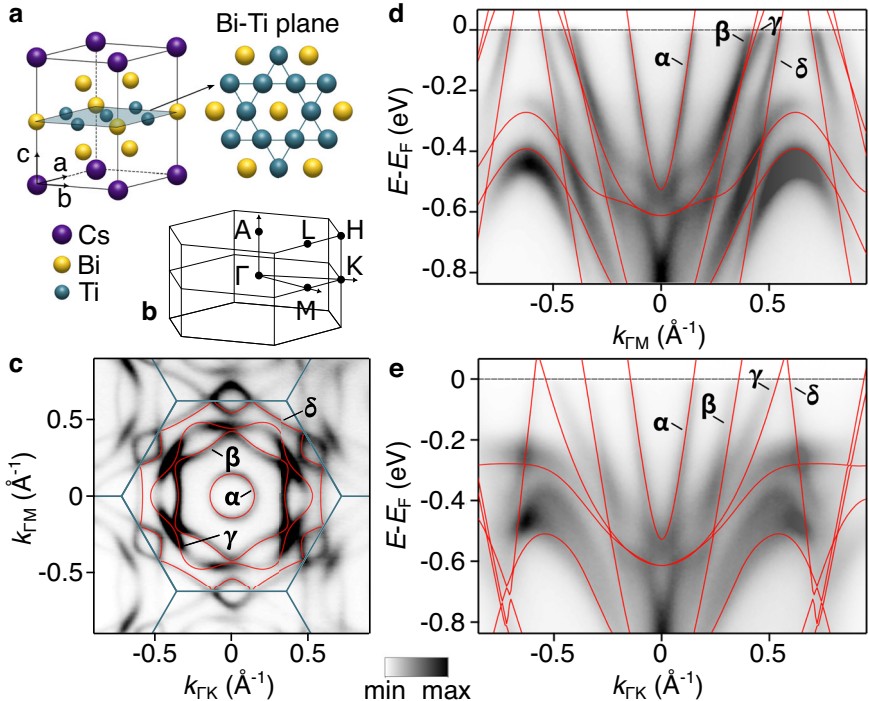

**Fig. 1 | Crystal and electronic structure of $CsTi_3Bi_5$. a** Cartoon of the unit cell structure. The shaded area marks the Bi-Ti plane where the Ti atoms form a kagome lattice. **b** Brillouin zone with high symmetry points. **c** Fermi surface collected at the Brillouin zone center with hν = 65 eV circularly polarised light (summing up right-

and left-handed polarisations, see full dataset in Supplementary Fig. 9). **d** Energy versus momentum dispersion along Γ-M and **e** Γ-K high-symmetry directions. Bulk, non-nematic first-principles band structures (red solid lines) are superimposed to the experimental data in **c**, **d** and **e**.

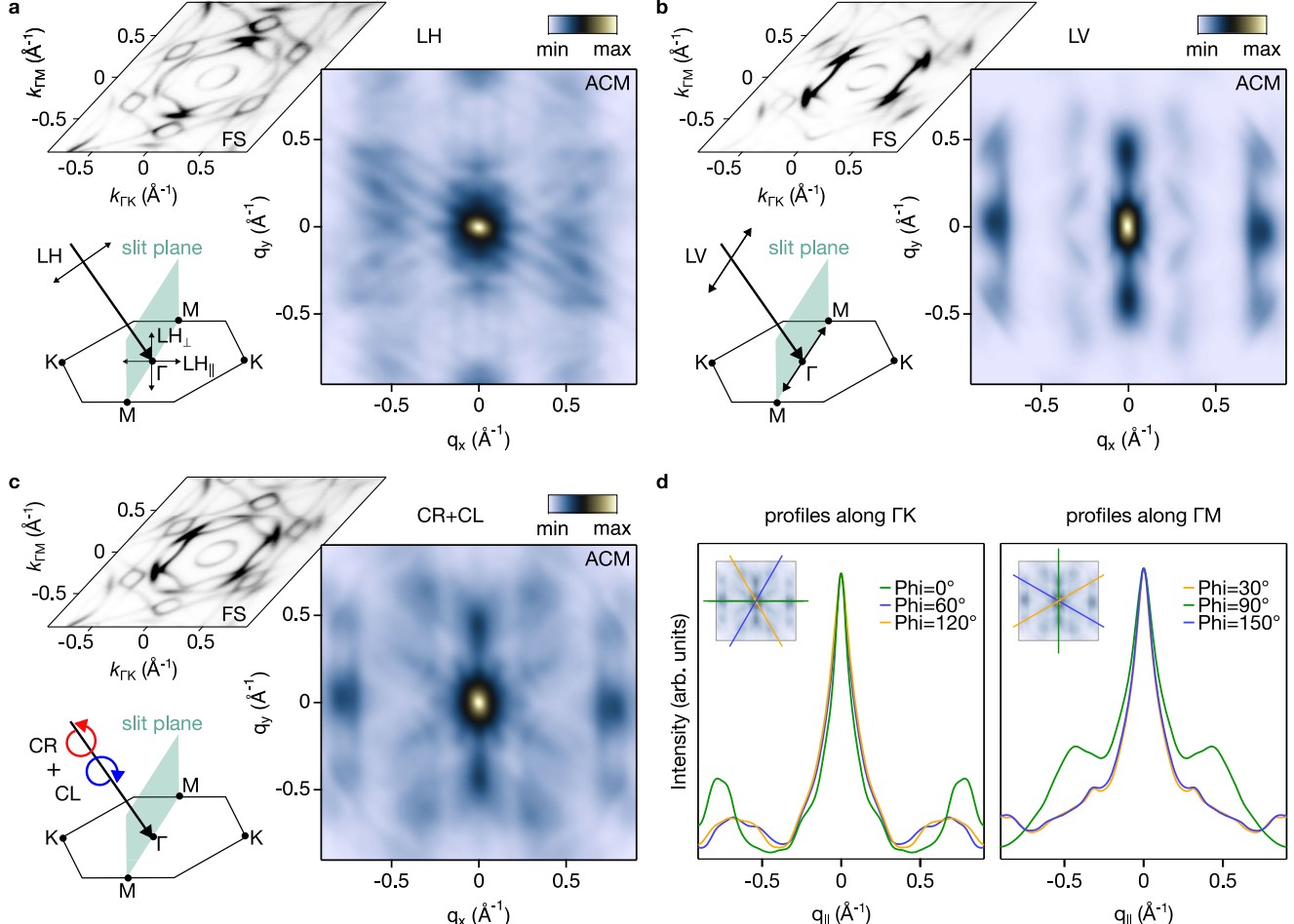

**Fig. 2 | Electronic nematicity of CsTi₃Bi₅ Fermi contour. a** Autocorrelation map (ACM) built from the Fermi surface contour measured at the Brillouin zone center with $h\nu = 65$ eV for linear horizontal polarised light and the analyser slit aligned along the $\Gamma$-M high-symmetry direction. The cartoon sketches the experimental geometry and vector projections. **b** Same as **a**, but for linear vertical polarised light, to rule out geometrical and matrix element effects. **c** Same as **a** and **b**, but for unpolarised light (summing up spectra collected with right- and left-handed circularly polarised light, see Supplementary Fig. 9). **d** Azimuthal profiles extracted from the ACM in **c**, supporting the reduced nematic $C_2$ symmetry. In all schematics shown in **a**–**c**, the mirror $xz$ plane contains the $\Gamma$-K direction. With $x$ and $y$ defining the sample surface and $z$ normal to it, LH has components along $x$ and $z$, while LV is oriented along $y$.

spectra collected using both circular-right and circular-left polarizations, ensuring comprehensive resolution of the principal electronic features. Additional spectra recorded with linear vertical and horizontal polarizations, along with varying photon energies, are provided in Supplementary Fig. 7 and Supplementary Fig. 8. Notably, the electronic band structure of CsTi₃Bi₅ reveals a pronounced absence of $k_z$ dependence for the dispersion at the energy of the Fermi level, with no significant variations detected upon photon energy adjustment (also consistent with the almost cylindrical ab initio Fermi surface discussed in Supplementary Note II), thereby underscoring its distinct two-dimensional character.

Significant spectroscopic variations emerge when the polarization vector of the incident light is modified. If the sample surface lies in the $xy$ plane, with the $z$ axis defined as perpendicular to this surface, then in our experimental geometry the light impinges on the sample at a 45° angle, i.e., the photon momentum is along the $x$ and $z$ directions. Although the crystal possesses several mirror planes, in this context we define the mirror plane as the one containing the incident light - namely, the $xz$ plane (in reciprocal space, this mirror plane contains the $\Gamma$-K direction.). Within this configuration, we refer to linear horizontal polarization as the electric field lying in the $xz$ plane, and to linear vertical polarization as the field oriented along the $y$ direction, perpendicular to the mirror plane. This is the geometry described in the

schematics of Fig. 2a, b. The photoemission intensity is strongly influenced by the polarization of the incident light, as it directly affects the transition probability between initial and final states. This probability is proportional to the square module of the dipole matrix elements as $|\langle \Psi_{fin}^k | \mathbf{A} \cdot \mathbf{p} | \Psi_{in}^k \rangle|^2$ and is non-zero only when the symmetry of $\mathbf{A} \cdot \mathbf{p} | \Psi_{in}^k \rangle$ coincides with that of the final state. For example, assuming the detector corresponds to a final state represented by a plane wave of even parity ($|+\rangle$), only transitions from initial states of even parity are allowed - specifically, $+|+\rangle$ and $-|-\rangle$. In contrast, transitions involving initial states of odd parity are forbidden ($\pm|\mp\rangle$), leading to a vanishing photoemission intensity[42,43]. Further discussion about the linear dichroism is reported in the Supplementary Information.

Understanding these polarization-dependent effects allows us to elucidate the orbital contributions to the electronic structure of CsTi₃Bi₅. This distinction is evident in the varying photoemission intensities observed for linear horizontal and vertical polarizations, as depicted in Fig. 2a,b, that highlight contrasting spectroscopic features. To enhance visualization, we also analyze the difference between the two polarizations (see Supplementary Note IV and Supplementary Fig. 4), providing insights into the relative contributions of even and odd orbitals with respect to the mirror plane.

Our findings reveal that the inner circular pocket ($\alpha$) centered at $\Gamma$ is predominantly composed of orbitals with even symmetry. The two

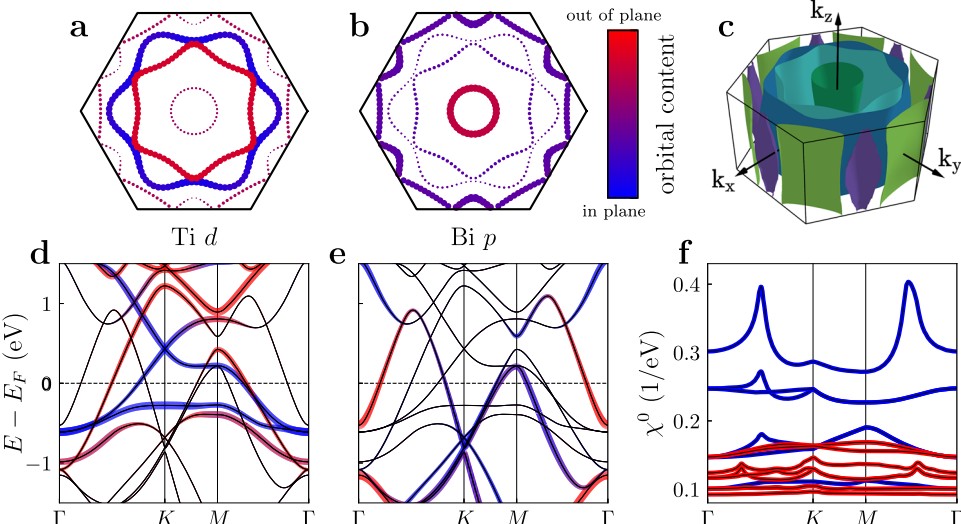

**Fig. 3 | Orbital polarisation of the Fermi sheets. a, b** Integrated orbital content on the $k_z = 0$ Fermi surface for the Ti $d$-orbitals and Bi $p$-orbitals, as indicated by dot size and color. The two inner Fermi pockets ($\beta$) and ($\gamma$) show clear orbital polarisation with respect to in-plane and out-of-plane alignment of the $d$-orbitals. **c** The three-dimensional Fermi surface shows little $k_z$ dispersion. **d, e** The orbital character of the full non-interacting dispersion for Ti $d$ and Bi $p$, revealing a low-lying Van Hove singularity that is exclusively supported by the in-plane $d$-orbitals. **f** This is directly reflected in the pronounced role of in-plane orbital combinations in the components of the orbital-resolved bare electronic susceptibility $\chi^0$ given by Eq. (1).

internal hexagonal Fermi surface sheets ($\beta$) and ($\gamma$), rotated by 30 degrees relative to one another, exhibit a substantial contribution from both even and odd orbitals. To further elucidate the relative contributions of even- and odd-symmetry orbitals while mitigating potential matrix element effects, we also performed light-polarization-dependent ARPES measurements across various experimental geometries, including rotations of the analyzer slit along the $\Gamma$-K direction (see Supplementary Fig. 11). The response of the two hexagonal Fermi surface sheets to this rotation is notably distinct. The intensity of the external ($\gamma$) sheet adapts in accordance with the sample orientation. In contrast, the internal ($\beta$) sheet demonstrates a shift in its spectral weight contribution, indicating a more complex orbital character. This behavior suggests that the external hexagonal ($\gamma$) sheet is primarily composed of in-plane orbitals, while the internal ($\beta$) sheet possesses a mixed orbital character. A solid support for this symmetry-based interpretation arises from the orbitally projected ab initio calculations shown in Fig. 3a and Supplementary Fig. 3, which unambiguously assign in-plane ($d_{xy}$, $d_{x^2-y^2}$) orbitals to the external ($\gamma$) sheet and a predominance of out-of-plane ($d_{xz}$, $d_{yz}$) orbitals to the internal one ($\beta$). Remarkably, this conclusion is consistent across multiple experimental configurations. Additionally, our use of varied light polarizations, different photon energies, and the absence of significant $k_z$ dispersion, further confirm that our findings are robust against photoemission matrix element effects.

We now focus on the nematic character of the system, which has been previously reported by STM experiments[31] to lead to a reduction of symmetry from $C_6$ to $C_2$. This alteration signifies that one of the three equivalent high-symmetry directions along both the $\Gamma$-K and $\Gamma$-M axes becomes distinct in terms of momentum length. To investigate this phenomenon, we perform Lorentzian fits to the momentum distribution curves at the Fermi level for inequivalent $\Gamma$-M directions. Our analysis reveals a subtle variation in the momentum positions of the bands that constitute the Fermi surface, primarily influenced by in-plane components (see Supplementary Fig. 10). However, the observed difference is minimal, and given the momentum resolution, it remains challenging to definitively attribute these variations to nematicity. Such effects are anticipated to be subtle, as highlighted in previous works[20,31].

Autocorrelation maps (ACMs) of the Fermi surface (see Fig. 2) instead yield valuable insights into the symmetries inherent in the

electronic structure, often elucidating even the most subtle features (Methods section for more details). CsTi$_3$Bi$_5$ serves as an ideal platform for this type of investigation, owing to the large extension of its nematic domains[31]. By generating ACM under varying light polarizations, we are able to uncover the nematic $C_2$ symmetry of the sample, independent of light polarization. This is clearly demonstrated in Fig. 2d, where profiles along the three $\Gamma$-M and $\Gamma$-K directions are presented. These profiles distinctly reveal that two directions remain equivalent, while the third exhibits a marked difference. The $q$-vector extracted from both the $\Gamma$-M and $\Gamma$-K directions in the ACM corresponds to scattering within the ($\gamma$) Fermi surface sheet, which is primarily composed of in-plane orbitals. Further corroboration of our findings comes from conducting the same analysis after rotating the sample (see Supplementary Fig. 11), or changing the photon energy (see Supplementary Fig. 12). Despite certain shape differences likely attributable to matrix element effects, two profiles consistently maintain equivalence while one consistently differs, thereby reinforcing the reduced $C_2$ symmetry of the system. Interestingly, the reduced symmetry persists throughout temperatures ranging from 16 K up to room temperature (RT), as we show in Supplementary Fig. 13. This result demonstrates that the nematic transition in CsTi$_3$Bi$_5$ is larger than RT, as no sign of transition was detected, consistently with a recent infrared spectroscopy study which also suggests the onset of nematicity in CsTi$_3$Bi$_5$ to happen above RT[44].

The subtle signatures of rotational symmetry breaking presented above hinder a definite determination of the nematic order parameter based solely on the experimental data. The latter, in fact, points towards the direction that the system is nematic: based on auto-correlation maps showing a $C_2$ symmetry and on electronic structure momentum distribution curves fitting, strong hints about this putative reduction are demonstrated. However, for substantial proof, it necessitates a detailed theoretical analysis on a realistic model to distinguish among the possible underlying microscopic origins[45]. In other "135" kagome compounds like AV$_3$Sb$_5$, nematicity is believed to emerge from a coupling between the different translation symmetry breaking orders at the inequivalent M points, that favors an imbalance between them to minimize the free energy within the symmetry broken phase[11,46] and is highly prone to external effects like in-plane strain and magnetic field[47]. The absence of a 2 × 2 superlattice modulation in

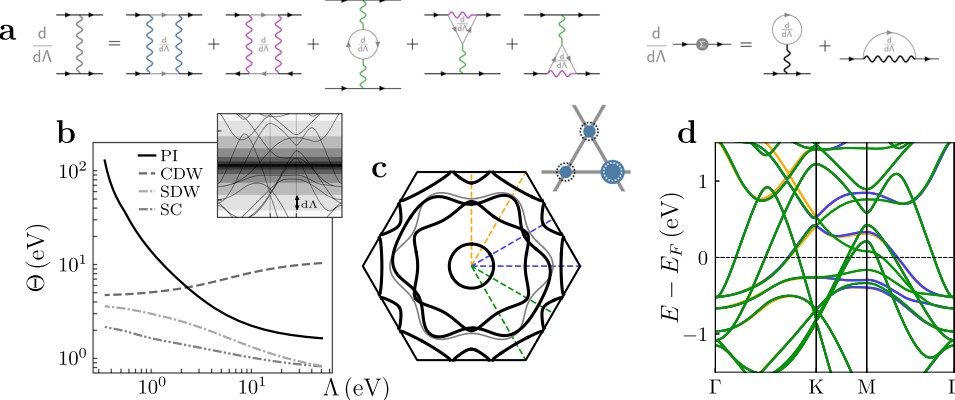

**Fig. 4 | Orbital selective nematicity from electronic correlations. a** FRG flow equations for the two-particle interaction vertex $V$ (left) and the self-energy $\Sigma$ (right). The different colored interaction lines indicate the transfer momenta characterizing the diagrammatic channels as particle-particle (P, blue), direct (D, green) and crossed (C, purple) particle-hole. Gray objects represent scale derivatives $d/d\Lambda$. **b** Eigenvalues of the scattering vertex $\Theta$ for superconductivity (SC), $2 \times 2$ charge density wave (CDW), magnetic order spin density wave (SDW) and Pomeranchuk instability (PI). The inset visualizes the successive integration of electronic modes in the energy range $|E| \in [\Lambda, \Lambda + d\Lambda]$ during each step of the FRG flow. **c** Nematic Fermi surface reconstruction. The $C_6$ symmetry of the original Fermi surface (gray) is lowered to $C_2$ in the symmetry broken phase (black) without any backfolding. The inset is a sketch of the corresponding charge distribution from the in-plane orbitals on the different kagome sublattice sites. **d** Band structure along the three formerly equivalent high-symmetry paths indicated in **c** in the nematic phase. Due to the broken $C_3$ symmetry by which $C_6$ is lowered to $C_2$, the Dirac cone is no longer pinned to the $K$ point.

the charge ordered phase of $CsTi_3Bi_5$ asks for a different explanation. Likewise, $CsTi_3Bi_5$ does not feature unstable phonon modes and the electron-phonon coupling is negligible[39]. Since phonons are not expected to drive a nematic instability in $d = 3$ dimensions[48] and hence seem not able to explain the accumulated experimental evidence, an exclusive electronic mechanism seems to drive the nematic transition in $CsTi_3Bi_5$.

To achieve a detailed understanding of the nematic transition and pin down its order parameter, we supplement the full ab initio dispersion (see Methods) with a electron-electron interactions between the Ti $d$-orbitals via a site-local Kanamori vertex and density-density interactions up to second-nearest-neighbor distance. We employ the FRG[49–52] in the timely truncated unity formulation (see refs. 53–57 and Methods) to analyse possible many-body instabilities of the system in an unbiased manner.

The large number of orbitals required to capture the low energy manifold is a common thread to any theoretical assessment of kagome metals. Hence, extracting the relevant degrees of freedom from the ab initio band structure is highly desirable to capture the kinetic theory in sophisticated many-body techniques. In the orbital-resolved band structure of Fig. 3d,e the three characteristic kagome bands featuring a flat band, a Dirac cone at K and VHss at M are clearly visible. As shown in Supplementary Fig. 4, the orbital parity of the model Fermi surface is corroborated by our ARPES linear dichroic data. This low energy kagome manifold is exclusively supported by the in-plane $d$-orbitals of Ti, while the out-of-plane Ti $d$ and Bi $p$ orbitals only contribute to bands with small spectral weight at the Fermi level. The orbital hierarchy is directly reflected in the static response of electronic states to quantum fluctuations of a certain orbital structure, that can be quantified by the orbital resolved bare susceptibility tensor

$$\chi^0_{o_1 o_2 o_3 o_4}(\mathbf{q}) = -\frac{1}{\beta} \sum_n \int_{BZ} \frac{d\mathbf{k}}{V_{BZ}} G_{o_2 o_4}(\mathbf{k}, \omega_n) G_{o_3 o_1}(\mathbf{k}+\mathbf{q}, \omega_n) + \text{h.c.} , \quad (1)$$

where $G_{o_1 o_2}(\mathbf{k}, \omega_n)$ is the single-particle propagator with momentum $\mathbf{k}$, fermionic Matsubara frequency $\omega_n$ and orbital quantum numbers $o_i$, and the integral over momentum and frequency is normalized by the Brillouin zone volume $V_{BZ}$ and inverse temperature $\beta$, respectively. The physical non-interacting susceptibilities, i.e., $\chi^0$ with $o_1 = o_2$, $o_3 = o_4$, are depicted in Fig. 3f. The effect of the isolated VHs of pure ($p$-type) sublattice character[1,58] in the vicinity of the Fermi level (Fig. 3d) is

directly apparent: while the peak in the non-interacting susceptibility is shifted from the Van Hove scattering vector M closer to $\Gamma$ due to the detuning of the chemical potential from the perfect Van Hove filling, the in-plane contributions to $\chi^0$ (blue) dominate over out-of-plane parts (red). Hence in-plane (anti-)screening processes play the decisive role in the determination of the system's ordering propensities and the emergence of a symmetry broken phase.

We therefore evaluate the FRG flow equations for the two-particle vertex (lhs. of Fig. 4a) with the in-plane orbitals as interacting subspace. We uncover a $Q = 0$ divergence in the interacting charge susceptibility (see also Supplementary Fig. 6), that corresponds to an intra-unit cell charge density wave order transforming under the two dimensional $E_2$ irreducible representation of the crystalline point group $P6/mmm$. In addition to the two-particle interaction, we also incorporate the static self-energy into the FRG flow (rhs. of Fig. 4a). Flowing into the symmetry broken phase[59] then yields the linear combination of order parameters minimizing the free energy. The resulting charge imbalance of the nematic state is depicted in the inset of Fig. 4c. Due to our choice for the interacting manifold, the reduction of $C_6$ symmetry down to $C_2$ is most prominently observed on the ($\gamma$) Fermi sheet in Fig. 4c, while the other sheets only exhibit a proximity induced warping due to hybridisation with the ($d_{xy}$, $d_{x^2-y^2}$) orbitals in good agreement with the ACMs of Fig. 2 and Supplementary Fig. 10.

To pin down the microscopic origin of this nematic charge order, we analyze the response of the system to the most prominent fluctuation channels on the kagome lattice by monitoring their instability scale throughout the FRG flow. In Fig. 4b, the fluctuation strength $\Theta$, extracted as the eigenvalue of the respective channel at a given wavevector $Q$, is depicted as a function of the energy cutoff $\Lambda$. We note that the $Q = 0$ charge order does already feature a sizable susceptibility compared to other orders involving translation symmetry breaking, i.e., with a finite $Q$ vector. This can be attributed to the fact that the long-range interaction supports charge imbalance between adjacent sites already at the mean-field level (see Supplementary Note V and Supplementary Fig. 6 for a comparison between mean-field and FRG phase diagrams). In the weak to intermediate coupling regime, this potential energy gain is usually dwarfed on the kagome lattice by the pronounced Van Hove scattering, that promotes a charge density with the nesting vector M[58,60] and a $2 \times 2$ supercell reconstruction. In most members of the 135 family, this charge modulation is indeed realized[11,16] due to the existence of low lying VHss of pure and mixed

sublattice type in the electronic spectrum[2,3]. Within the FRG formulation, this effect can be directly observed via a strong enhancement of the M-point charge susceptibility as the cutoff approaches the Fermi level[60,61]. In the case of $CsTi_3Bi_5$, there is only a single p-type VHs close to the Fermi level and the chemical potential of the system is detuned from Van Hove filling by $\approx 0.22$ eV, which substantially reduces the effect of Van Hove nesting (see Supplemental Note V for further details). With the accompanied sublattice interference still active[1], the accessible phase space for on-site scattering in the Van Hove channel remains limited and the local repulsion can be dynamically overscreened by long-range interactions, which suppresses the formation of local moments and the emergence of magnetic orders.

In addition, the potential energy gain in the geometrically frustrated kagome geometry is also supplemented by a kinetic energy gain. The charge imbalance pushes two VHss in the quasi-particle band structure up in energy, while the third one moves closer to the Fermi level, as shown in Fig. 4d. Thereby, the spectral weight is more efficiently moved away from the Fermi level than by the opening of a charge ordered gap at the VHs. This combined effect of electronic kinematics and interactions leads to a spontaneous breaking of rotational symmetry by implying non-uniform sublattice occupation without enlarging the unit cell.

In this respect, $CsTi_3Bi_5$ provides a rare material realisation of a d-wave Pomeranchuk instability in its original sense, namely the spontaneous breaking of pointgroup symmetry by correlation effects resulting in a smooth deformation of the Fermi surface[33,62]. This is directly reflected in the divergence of the associated scattering vertex in Fig. 4b at the critical scale. In the emergence of a PI, the non-trivial sublattice structure of the kagome lattice supplies additional degree of freedom and finite angular momentum channels, that allow to circumvent the constraints for PI in conventional Fermi liquid theory: if there is only one electron species present in the system, a PI can only manifest as bond or current order[63], whose emergence is often hampered by charge and spin conservation rules[64,65]. While these states accumulate a finite angular momentum in the relative coordinate between the constituents of the particle-hole pair, the obtained d-wave PI state in $CsTi_3Bi_5$ is characterized by a $l = 2$ quantum number in the total angular momentum of the charge order parameter within the unit cell. Therefore, a charge accumulation on one site and orbital can be compensated by a partial depletion on a different site, as apparent from Fig. 4c.

## Discussion

Our results differ from the current opinion that the nematic charge order in $CsTi_3Bi_5$ is realised by a bond-type order with zero total momentum[20,31]. That assessment was based on the observation of a momentum-dependent nature of the nematic Fermi surface distortion, which was incorrectly claimed to be unexplainable by a simple charge imbalance between the unit cell sites[20]. Diagrammatic calculations also suggested a bond-type order[34]. It is however important to highlight that this previous study includes only a subset of the functional renormalization group (FRG) diagrams, leading to a pronounced bias towards non-local charge orders and pronounced features from VHss. Our unbiased FRG analysis reveals that these additional diagrams indeed favor the emergence of a site-type order in the $E_2$ irreducible representation and disfavor odd-parity bond order states[64,65]. Consistency between the experimentally observed Fermi surface warping and a charge imbalance between different unit cell sites is achieved by the orbital structure of the PI, that induces a non-trivial momentum space structure of the nematic quasi-particle band structure via the orbital-to-band transformation. The resulting Fermi surface reconstructions for both site- and bond-type order differ only sightly, underscoring the subtle complexities involved in characterizing intraunit cell orders and the necessity of a synergetic approach via spectroscopic signatures like ARPES and detailed theoretical modelling.

In this study, we have elucidated the origin of the nematic character in $CsTi_3Bi_5$ by correlating high-resolution light-polarization angle-resolved photoemission spectroscopy data with ab initio-based many-body theoretical predictions. Our findings suggest that the nematicity is predominantly driven by electronic correlations through a complex orbital-selective mechanism and detuning of the chemical potential away from Van Hove filling. This work not only contributes to our understanding of nematic behavior within the 135 family of kagome materials but also provides critical insights into its underlying origins and manifestations. By establishing this connection, we paved the way for further exploration of electronic correlations and their implications in complex quantum materials.

On a broader perspective, our work suggests a generic propensity of PIs in systems with a non-trivial unit cell based on the frustrated hexagonal lattice, beyond $CsTi_3Bi_5$. This seems to be witnessed by observations in related materials: In regard to other Ti-based kagome systems, previous studies on the sister compound $RbTi_3Bi_5$ employed ARPES measurements to investigate aspects of nematicity[20,32]. Despite their almost identical electronic structure from DFT calculations, their experimental signatures vastly differ[66,67]. In particular, optical probes seem to suggest the nematic transition in $RbTi_3Bi_5$ to set in around 200 K while our work provides evidence for nematicity above room temperature in $CsTi_3Bi_5$[44]. This discrepancy is complemented by different orbital characters of the nematic band reconstruction: while in our work on $CsTi_3Bi_5$ we have established a predominant role of Ti $d_{xy, x^2-y^2}$ orbitals in promoting a Pomeranchuk instability, in $RbTi_3Bi_5$ $p_{x,y}$ and $d_{xz,yz}$ orbitals seem to contribute substantially to the formation of nematicity, along with significant interband scattering, likely originating from the second hexagonal Fermi surface sheet and pockets at the M point. This calls for further theoretical studies on $RbTi_3Bi_5$ to pin down the microscopic cause for these apparent discrepancies. Beyond the family of 135, in the kagome bilayer material $ScV_6Sn_6$, a stripe-like nematic order was recently identified by STM measurements[21]. Due to the complexity of the electronic structure, the microscopic origin of this PI could not be identified, but was speculated to arise from Van Hove scattering effects[68]. The reduced complexity of $CsTi_3Bi_5$ allowed to pinpoint the physical origin of a nematic instability using rigorous many-body numerical techniques based on ab initio calculations. In conclusions, our findings can serve as a guiding framework for understanding nematicity in other kagome metals, where multiple VHss near the Fermi level may cooperate to drive symmetry-breaking instabilities.

## Methods

### Bulk single-crystal synthesis

Single crystals of $CsTi_3Bi_5$ were grown using a conventional flux-based growth technique, as described previously[69]. The elemental ingredients for synthesis included Cs (liquid, Alfa 99.98%), Ti (powder, Alfa 99.9%), and Bi (shot, Alfa 99.999%). These materials were loaded into a tungsten carbide milling vial in a stoichiometric ratio of 1:1:6 and milled for an hour under an argon atmosphere to obtain precursor powder. After milling, the precursor powder was transferred into an alumina crucible and sealed in a separate stainless-steel tube. The samples were heated at 900°C for 10 h and then cooled at 3°C/hr to 500°C. Once the growth period was over, the tube was broken inside a glove box filled with Ar-gas. The shiny plate-like single crystals were separated gently, and stored inside a box filled with an inert gas atmosphere. As also shown in previous studies[31], the single domains are extremely large in $CsTi_3Bi_5$ compared to those of the vanadium-based sister compounds. This aspect permits the extraction and analysis of autocorrelation maps from the ARPES spectra of Fermi surface.

### ARPES experiments

The samples were cleaved in ultrahigh vacuum (UHV) at the pressure of $1 \times 10^{-10}$ mbar. The ARPES data were acquired at the CASSIOPEE end

station of the synchrotron radiation source SOLEIL (Paris, France). The energy and momentum resolutions were better than 10 meV and 0.018 Å$^{-1}$, respectively. The temperature of the measurements was kept constant throughout the data acquisitions (15 K), except for the temperature dependent ARPES measurements reported in Supplementary Fig. 13, for which a scan up to room temperature was done. The Fermi surfaces were collected by rotating the angle orthogonal to the analyser slit, keeping the samples in the center of rotation. To exclude possible matrix elements effects, several light polarizations, geometries, and photon energies were used for the data acquisition, yielding always consisting results.

## Autocorrelation analysis of ARPES data

To analyze symmetry breaking and identify characteristic scattering vectors in the Fermi surface, we employed the autocorrelation method (ACM) on angle-resolved photoemission spectroscopy (ARPES) intensity maps. This approach allows for the extraction of periodic features in momentum space directly from within a single Brillouin zone (BZ), without requiring data acquisition over multiple repeated BZs.

The autocorrelation function $A(\mathbf{q})$ is defined as the self-convolution of the ARPES intensity $I(\mathbf{k})$ in momentum space:

$$A(\mathbf{q}) = \int_{BZ} I(\mathbf{k}) \cdot I(\mathbf{k} + \mathbf{q}) \, d\mathbf{k}, \tag{2}$$

where $\mathbf{k}$ and $\mathbf{q}$ are momentum vectors. The integral is performed over the first Brillouin zone, where ARPES measurements are typically confined. Peaks in $A(\mathbf{q})$ correspond to dominant scattering vectors connecting high-intensity regions of the Fermi surface. These vectors are often associated with quasiparticle interference (QPI) patterns, similar to those observed in Fourier-transformed scanning tunneling microscopy (STM) data.

Compared to a direct Fourier transform (FFT) of the raw Fermi surface, the ACM offers several practical and methodological advantages:

- Single-zone applicability: Unlike FFT, which ideally requires sampling across multiple Brillouin zones to reveal periodicity, ACM operates entirely within a single BZ. This is essential in ARPES, where the experimental constraints (angular resolution, matrix element effects, photon energy limitations) make extended-zone mapping impractical.
- Preservation of matrix element conditions: Since all momentum points used in ACM lie within the same BZ, the photon-induced matrix element effects remain consistent, thereby avoiding artefacts that could arise in FFT from variable photoemission intensities across zones.
- Enhanced signal-to-noise ratio: By construction, the autocorrelation enhances symmetric and repetitive structures while suppressing random noise and asymmetries. This makes it particularly effective for detecting weak periodic modulations in complex or reconstructed Fermi surfaces.
- Direct physical interpretation: Peaks in the autocorrelation map have a straightforward interpretation as representative scattering vectors. In systems exhibiting nematicity, charge order, or other symmetry-breaking phenomena, these features may correspond to nesting vectors or interaction-driven instabilities.

While FFT and ACM are formally related, both probing periodic structures in the data, the ACM is better suited for ARPES-derived momentum maps, especially when high-resolution measurements are confined to a single BZ. This method is increasingly being adapted for momentum-space techniques such as ARPES, offering a powerful tool for characterizing symmetry-lowering phenomena in correlated electron systems.

## First-principles calculations

Bulk electronic structure calculations were performed using the full-potential local-orbital (FPLO) code (v.21.00-61)[70]. The unit cell has lattice constants of 5.82709 Å, 5.82709 Å, and 9.93612 Å. The exchange-correlation energy was parametrized within the local density approximation, following the Perdew-Wang 92 formulation[71]. A $12 \times 12 \times 12$ $k$-grid was used to sample the Brillouin Zone, and the tetrahedron method was employed for integration. Calculations were performed in both the full-relativistic and non-relativistic frameworks.

A subsequent 36 bands Wannier functions model[72,73], with symmetries accurately implemented, was constructed considering projections onto the following states: Caesium $6s$; Titanium $3d_{z^2}$, $3d_{xz}$, $3d_{yz}$, $3d_{x^2-y^2}$, $3d_{xy}$; and Bismuth $6p_z$, $6p_x$, $6p_y$, $6s$. The Wannier states correspond to real spherical harmonics with a global quantization axis independent of the position within the kagome unit cell. The atomic orbital content of band $n$ and momentum $\mathbf{k}$ is hence inferred from the electronic eigenstates $\psi_{no}(\mathbf{k})$ as $\sum_i |\psi_{no_i}(\mathbf{k})|^2$ where we sum over the same orbitals of equivalent atoms. From the analysis of the Fermi surface weights in the Supplementary Information it becomes directly evident, that only Ti $3d_{xz}$, $3d_{yz}$, $3d_{x^2-y^2}$, $3d_{xy}$ and Bismuth $6p_z$, $6p_x$, $6p_y$ contribute sizable to the Fermi surface states. We obtain the Fermi surface weights of the in- and out-of-plane orbitals in Fig. 3 by summing up the contributions of the Ti $3d_{x^2-y^2}$, $3d_{xy}$; Bi $6p_x$, $6p_y$ and Ti $3d_{z^2}$, $3d_{xz}$, $3d_{yz}$; Bi $6p_z$, respectively.

Surface electronic structure calculations were performed using the Vienna Ab initio Simulation Package (VASP)[74,75], using the projector augmented wave (PAW) method[76]. Exchange and correlation effects have been handled using the generalized gradient approximation (GGA)[77-79] within the Perdew-Burke-Ernzerhof (PBE) approach[80]. The truncation of the basis set was set by a plane-wave cutoff of 500 eV and a $12 \times 12 \times 1$ $k$-grid was used.

## Many-body calculations

The bare susceptibility presented in Fig. 3f was calculated via Equation (1) in the famework presented in ref. 81 with a $600 \times 600 \times 1$ integration grid in the Brillouin zone at an inverse temperature of $\beta = 100$ 1/eV. The different lines correspond to all possible orbital combinations of the physical susceptibility $\chi^0_{o_1 o_1 o_2 o_2}(\mathbf{q})$ of the Ti $d$-orbitals, where blue (red) coloring indicates $o_1, o_2 \in \{d_{xy}, d_{x^2-y^2}\}$ ($o_1, o_2 \in \{d_{xz}, d_{yz}, d_{z^2}\}$). The mixed contributions and also contributions from the Bi $p$ and Cs $s$-orbitals are all smaller and hence not shown.

To analyse the symmetry breaking propensies of $CsTi_3Bi_5$, we employed the functional renormalization group (FRG) in the truncated unity formalism (and its static approximation using a sharp frequency cutoff) as implemented in the divERGe library[57]. We restrict the interacting subspace to the in total six in-plane Kagome $d$-orbitals and employ a formfactor cutoff of 1.1 in-plane lattice constants. This corresponds to real space form-factors up to fourth nearest neighbour, that have shown to produce converged results on the kagome Hubbard model[59]. The flow is started at $\Lambda_0 = 500$ eV and integrated logarithmically with step size $d\Lambda = -0.05\Lambda$ (using the adaptive Euler integrator shipped with divERGe). We classify the vertex as divergent when a single element surpasses $V_{\max} = 50$ eV. Even though effects of the out-of-plane dispersion on our FRG results are very weak (Supplementary Note II), we evaluate the scattering vertex in the respective channels on a regular $24 \times 24 \times 6$ mesh in the full 3D Brillouin zone, with an additional refinement of $12 \times 12 \times 5$ (per coarse mesh point, i.e., $(24 \times 12) \times (24 \times 12) \times (6 \times 5)$ points in total) for the integration of the internal loops in the flow equations. With the given k-mesh we are able to resolve energy scales down to $10^{-3}$ eV, which is well below the critical scales in our calculations.

The FRG interpolates between the full interacting model and an effective low energy theory close the Fermi level by adding a regulator $\Lambda$ to the bare Green's function, that recovers the non-interaction limit as $\Lambda \to \infty$, and separates the electronic states into fast and slow

modes[50,51]. As the cutoff $\Lambda$ is successively lowered, the fast modes are integrated out and provide the screening background for the effective interaction of the remaining degrees of freedom via the mutual electron-electron interactions between fast and slow modes. The change of the effective $n$ − particle interaction as a function of the cutoff is best expressed in an infinite hierarchy of coupled differential flow equations for the $2n$ − particle vertex, that is conventionally truncated within one loop order, i.e., at second order in the interactions, and gives way to the FRG flow equations depicted in Fig. 4a. By monitoring the evolution of the vertex throughout the flow, the FRG provides a transparent way to pinpoint the origin of a symmetry breaking transition. Its implicit diagrammatic resummation is unbiased in the sense that it includes all ladder diagrams of the particle-particle (P), direct particle-hole (D), and crossed particle-hole (C) channel, as well the leading order vertex corrections between them. This cross talk between the different diagrammatic channels proves quintessential to approach cooperative and competing instability phenomena characteristic for the highly frustrated kagome lattice[14,59].

We mimic the mutual electron-electron interaction within the Ti $d$-orbital manifold at the bare level by the two particle vertex

$$
\begin{aligned}
\hat{H}_I = & U \sum_{io} \hat{n}_{io\uparrow} \hat{n}_{io\downarrow} + U' \sum_{io_1 < o_2 \sigma\sigma'} \hat{n}_{io_1\sigma} \hat{n}_{io_2\sigma'} \\
& + J \sum_{io_1 < o_2 \sigma\sigma'} \hat{c}^\dagger_{io_1\sigma} \hat{c}^\dagger_{io_2\sigma'} \hat{c}_{io_1\sigma'} \hat{c}_{io_2\sigma} + J' \sum_{io_1 \neq o_2} \hat{c}^\dagger_{io_1\uparrow} \hat{c}^\dagger_{io_1\downarrow} \hat{c}_{io_2\downarrow} \hat{c}_{io_2\uparrow} \\
& + V_1 \sum_{\langle i,j \rangle, o_1 o_2 \sigma\sigma'} \hat{n}_{jo_1\sigma} \hat{n}_{io_2\sigma'} + V_2 \sum_{\langle\langle i,j \rangle\rangle, o_1 o_2 \sigma\sigma'} \hat{n}_{jo_1\sigma} \hat{n}_{io_2\sigma'} ,
\end{aligned}
\tag{3}
$$

where $\hat{n}_{io\sigma} = \hat{c}^\dagger_{io\sigma} \hat{c}_{io\sigma}$ is the fermionic number operator on site $i$ with orbital $o$ and spin $\sigma$. $\langle i,j \rangle$ and $\langle\langle i,j \rangle\rangle$ indicate a summation over nearest neighbor (NN) and second nearest neighbor (NNN) sites respectively.

In accordance with the dominant role of the in-plane Ti $d$-orbitals in the low-energy kinematics described above (Fig. 3d,e), we choose the interacting manifold as the in-plane Ti $d$-orbitals, while all other orbitals only provide the screening background for the two particle interaction in this channel. We have confirmed the validity of our results via calculations with an interacting manifold containing the full $d$-shell (Supplementary Note III).

We use the universal Kanamori relations for the onsite interaction tensor $U = U' + 2J$ and $J = J'$[82] with $J = 0.8$ eV and set $U = 4$ eV, $V_1 = 1.5$ eV, and $V_2 = 0.5$ eV for all data provided in the main paper by employing the universal decay behavior for density-density interactions in the 135 family obtained from constrained RPA calculations[83]. All our results are robust against moderate changes of the interaction profile as shown in the Supplementary Information.

The FRG flow is stopped when encountering a divergence of the effective vertex. This indicates a phase transition and we can evaluate the leading instability by solving the linearized gap equation in the three diagrammatic channels. This amounts to diagonalizing the interaction vertex in the basis of composite two fermion operators. While the eigenvalue $\Theta$ gives the scattering amplitude of the respective fluctuation, the associated transfer momentum $\mathbf{q}$ and the eigenstate determine the spatial structure of the order parameter. For further details we refer to the standard review articles[50,51]. Here we just note, that we consider all possible transfer momenta contained in the momentum discretization of the BZ as well as all possible secondary momentum dependence of the order parameter captured within the employed formfactor shells. In Fig. 4b we exemplarily display the maximum eigenvalues of the pairing channels at $\mathbf{q} = 0$ (SC), crossed particle-hole channel at $\mathbf{q} = M$ (SDW), and the direct particle-hole channel at $\mathbf{q} = 0$ (PI) and $\mathbf{q} = M$ (CDW) without restricting the functional form of the eigenvector at these momentum space points. Hence, the CDW e.g., also contains contributions from bond order fluctuations.

## Data availability

The raw data generated in this study have been deposited in the Zenodo database under accession code https://zenodo.org/records/17641746.

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

## Acknowledgements

We acknowledge SOLEIL for provision of synchrotron radiation facilities under proposal No. 20231813. M.D., R.T. and G.S. are supported by the Deutsche Forschungsgemeinschaft (DFG, German Research Foundation) through Project-ID 258499086 - SFB 1170 and through the Würzburg-Dresden Cluster of Excellence on Complexity and Topology in Quantum Matter - ct.qmat Project-ID 390858490 - EXC 2147. F.M. is grateful for the project funded by the European Union - NextGenerationEU, M4C2, within the PNRR project NFFA-DI, CUP B53C22004310006, IR0000015. M.D. is grateful for support from a Ph.D. scholarship of the Studienstiftung des deutschen Volkes. A.C. acknowledges support from PNRR MUR project PE0000023-NQSTI. A.C., R.T., G.S. and D.D.S. acknowledge the Gauss Center for Supercomputing e.V. (https://www.gauss-centre.eu) for funding this project by providing computing time on the GCS Supercomputer SuperMUC-NG at Leibniz Supercomputing Center (https://www.lrz.de). M.D., L.K. and R.T. are grateful for HPC resources provided by the Erlangen National High Performance Computing Center (NHR@FAU) of the Friedrich-Alexander-Universität Erlangen-Nürnberg (FAU), that were used for the FRG calculations. NHR funding is provided by federal and Bavarian state authorities. NHR@FAU hardware is partially funded by the DFG - 440719683. I.Z. gratefully acknowledges the support from the National Science Foundation (NSF), Division of Materials Research 2216080. S.D.W. and G.P. gratefully acknowledge support via the UC Santa Barbara NSF Quantum Foundry funded via the Q-AMASE-i program under award DMR-1906325. Work by B.R.O. was supported by the U.S. Department of Energy (DOE), Office of Science, Basic Energy Sciences (BES), Materials Sciences and Engineering Division. D.O. and R.C. acknowledge support by the Air Force Office of Scientific Research under grant FA9550-22-1-0432. J.A.M acknowledges funding from DanScatt (7129-00018B). We thank C.A. Baum and H. Hohmann for valuable feedback on the FRG calculations. We thank V. Brouet for valuable discussions.

## Author contributions

C.B., F.M., F.B., P.L.F., A.D.V., M.Z. conducted the ARPES experiments and C.B. and F.M. analysed the experimental data in consultation with T.J., H.C.T., P.T., J.A.M, J.W.W., D.O., R.C. and I.Z. M.D., L.K. and A.C. carried out the theoretical analysis in consultation with D.D.S., G.S. and R.T. S.D.W., B.O. and G.P. performed the crystal synthesis and structural characterization. F.M., C.B., M.D. and D.D.S. wrote the paper with contributions from all authors. All authors discussed the results, interpretation and conclusion.

## Funding

## Competing interests

The authors declare no competing interests.
