## [Transparent Peer Review file · Nature Communications]

Pomeranchuk instability from electronic correlations in CsTi₃Bi₅ kagome metal

Corresponding Author: Professor Domenico Di Sante

Version 0:

Reviewer comments:

Reviewer #1

(Remarks to the Author)

The manuscript of Bigi et al shows ARPES results together with theoretical study on the origin of the electronic structure showing nematicity in CsTi₃Bi₅. I think the work has been conducted nicely and the paper is interesting. Basically, it can be said that this is a follow up of what is reported in Nature Physics 19, 1591 (2023) using ARPES in terms of experiment. The theoretical part is the novel part, suggesting a new origin of the observed Pomeranchuk instability. I suggest the authors to consider the following points before I can recommend it for publication in Nat. Comm.

1. The authors refer to the autocorrelation maps (ACMs) of the Fermi surface to demonstrate the nematicity. I think it is important, but I would like the authors to first perform FFT on the raw data of the Fermi surface first before discussing the ACMs; it should be similar to the analysis of Nature Physics 19, 1591 (2023), and although it maybe not so clear, one should be able to recognize the symmetry lowering at this stage. If this is not the case, I cannot really trust the ACMs. The authors also should add an explanation of the ACMs. How it is derived (the equation), the physical meaning, etc.
2. In Figs. 1 d and e, the experimental data is compared to the DFT calculations. The overall consistency is good, but near the Gamma point at -0.6 to -0.8 eV, there are some deviations. The authors should discuss this point.
3. The authors should explain what is the temperature that the Pomeranchuk instability occurs in the experiment, and how can it be explained from the present theory.

Reviewer #2

(Remarks to the Author)

This paper is composed of two parts. The first half is the experimental results of polarization-dependent ARPES performed on CsTi₃Bi₅. As a result, the authors found that of the four Fermi sheets (α , β , γ , δ), the α and δ have the character of out-of-plane orbitals, and the γ has the character of dx_y, dx^2-y^2 . It was proposed that the β has dx_z, yz character. It was also confirmed that the rotational symmetry breaking phase (C2) proposed for this material occurs on the γ Fermi surface. In the second half, which is theoretical, the origin of the C2 state was proposed based on many-body calculation techniques (fRG). As a result, a Pomeranchuk state in which an imbalance in electron density occurs was proposed.

In various Kagome metal systems, it is well known that orbital properties are important to understand a lot of phase transitions. Thus, the present attempt to detect the orbital dependence of the Fermi surface using polarization-dependent ARPES can be evaluated to a certain extent as valuable. On the other hand, there are many concerns where the theoretical logic for interpreting the experimental results is unclear or explanations are missing. Throughout the manuscript, the authors' conclusions are often exaggerated and leaps are made in comparison to the experimental results actually obtained. Therefore, this manuscript does not reach the level required for publication in Nature Communications. I recommend that the gap between them be properly filled and that the paper be considered for publication in more specialized journal.

The main concerns are as follows.

1. The authors concluded that the following in-plane/out-of plane orbital dependence can be obtained from the Figure 2 (a, c). However, there appears to be a jump between their conclusion and the experimental results.

"Our findings reveal that the inner circular pocket (α) centered at Γ is predominantly composed of orbitals with out of plane symmetry. Additionally, the outermost hexagonal Fermi surface sheet (δ), forming the pockets around both the M and K points, exhibits similar characteristics. In contrast, the two internal hexagonal Fermi surface sheets (β) and (γ), rotated by 30 degrees relative to one another, exhibit a substantial contribution from in-plane orbitals"

This is because, from the linear polarized ARPES experiment, we estimate that the parity with respect to the mirror plane can be identified, then wouldn't the experimental results in Figure 2 (a,c) only reveal the parity with respect to the mirror plane? It seems not self-evident how the authors' conclusion that

"We have discovered that the α and δ planes have the character of interplane orbitals, and the β and γ planes have the character of in-plane orbitals " is derived from this. The clear relationship between the light-polarization (LH and LV) and orbital symmetry should be formulated in the main text.

2. In relation to the above, even if we consider a dxy orbital on kagome lattice, there are three sublattices. Then, three orbital do not form 90 degrees, so it seems non-trivial how the symmetry argument correspond to the results of this study, and a proper explanation is required.

3. Furthermore, the authors conclude that the γ plane is dxy, dx^2-y^2 and the β plane is dxz, yz , based on the results of changing the slit direction from the original Γ -M to the Γ -K direction, but it is difficult to understand because they do not quote the figure numbers in the text. The actual experimental results are presumed to be extended date Fig.5, but shouldn't this figure be included since it is important for identifying the orbit? In addition, a discussion of the symmetry argument to derive these results should be included using equations, etc.

" we also performed light-polarization-dependent ARPES measurements across various experimental geometries, including rotations of the analyzer slit along the Γ -K direction. The response of the two hexagonal Fermi surface sheets to this rotation is notably distinct."

4. In identifying the orbitals of the β and γ planes, it is necessary to clarify the starting point for the orbital candidate trajectory and how the trajectory was narrowed down based on the each h experimental results (with appropriate reference to the figure). For example, why do p orbitals, s, $d_{3z^2-r^2}$ were excluded?

5. From the results, shouldn't they include information on which orbitals are more prevalent at which wave number k on which Fermi surface? They concluded that there are two types of orbitals, dxy and dx^2-y^2 for the γ and dxz and yz for the β , but are the orbital weight the same along k, or do they change ? Including this result would also be helpful for comparison with first-principles calculations."

6. Between extended date Fig. 5(d) and Fig. 2(d), the asymmetry of C2 appears to be more clearly suppressed in Fig. 5(d). What is the reason for this?

7. The definition of the orbital basis should be explained. The authors call the orbitals in this study dxz , dyz , etc., but most readers who are not familiar with this material will not understand which orbital is which in real space. The definition of the orbital should be illustrated.

Regarding the theory part, the conclusion is that a Pomeranchuk instability occurs based on a certain model parameter (initial value of the interaction U,V1,V2 and temperature). However, since there is almost no information on the robustness of the results, it is impossible to judge how reasonable the results are from the manuscript as it is, and doubts from many readers cannot be dispelled. It should be fundamentally revised.

8.The authors conclude that such Pomeranchuk instability appear due to the inclusion of many-body effects by fRG compared to the mean-field method, but this conclusion is not immediately convincing because the results of solving the same model at the mean-field approximation level are not presented. The authors should present the mean-field results of solving the same model at the same temperature as fRG with the same magnitude of interactions (Of course, they should verify the robustness by changing U,V1,V2.). In particular, can they say that the Pomeranchuk instability (PI) does not appear in the mean-field approximation even if the magnitudes of the long-range Coulomb interactions V1 and V2 become somewhat large? If PI appears within the mean-field model, the validity of the magnitude of the interactions V1V2 in this study must be explained.

9. It is known that low-temperature calculations are difficult in such mesh fRG because it is generally difficult to obtain fine k-meshes. However, was this calculation performed at a temperature that is consistent with the transition temperature of about 100K observed in experiments? If the answer is Yes, it should be discussed whether this is a reliable region with the present number of k-mesh. If the answer is No, is there a guarantee that the same PI will be dominant at a low temperature of around 100K? These temperature dependence needs to be discussed.

10. In Fig. 4(b), the divergent growth of two-particle interaction vertex does not stop as a function of $\Lambda \sim 10000K$. On the other hand, what would happen if they used parameters U,V that can be renormalized up to around $\Lambda = 100K$. Such parameter region would be more consistent with the experiment."

11. In relation to the above, what is the estimate of the C2 transition temperature based on the fRG theory?

12. I Since the main text only provides results for non-interacting susceptibility in Fig.3 (f), in order to compare with the fRG results, the results for susceptibility analyzed at the mean-field level using the same model as fRG (J = 0.8 eV, U = 4, V1 =

1.5, $V_2 = 0.5$) should be presented and compared.

13. The validity of the results cannot be judged because there is no definition of PI, CDW, SDW, and SC in Fig. 4(b). The definition should be written in terms of wave number dependence and orbital dependence. Presumably, using the center of mass momentum Q of the particle-hole pair, the PI is $Q = 0$ CDW and the CDW is $Q \neq 0$ CDW. In addition, when writing Fig. 4(b), is a specific wave number $Q^* \neq 0$ fixed? In that case, has the possibility of the CDW for other Q being larger been eliminated? Also, does this fRG formalism take into account the bond-type CDW proposed in the Vanadium-based kagome systems? The k -dependence of the obtained vertex should be explained. Also, where did the spin channel for $Q = 0$ disappear to?

14. The definitions of the terms in-plane and out-of-plane in the color bar in Fig. 3(a) are not clear, making it difficult to judge whether the theoretical model is consistent with the experimental results. Which orbital is in-plane dx or dx^2-y^2 ? out-of-plane dxz or yz ? It should be specifically written. Also, does the orbital's character differ depending on the k ? At the same time, it should be shown whether their k -dependence is consistent with ARPES data. Based on that, does the even/odd nature of this orbital match the experimental results?

15. For the theoretical analysis, they should compare the phase diagram by RPA and fRG by changing the values U, V_1, V_2 for showing the robustness of their main result. It is expected that PI disappears when V_1, V_2 are zero or quite small even if they performed fRG. What is the critical value of V_1 and V_2 ?

16. Shouldn't the results of changing the filling also be discussed using the fRG phase diagram? Since they conclude that the distance of the van Hove singularity (vHS) from the Fermi surface is important for the emergence of the PI, it seems crucial to demonstrate that when the vHS gets closer to the Fermi surface, a different instability switches on."

Anyway, the paper is not self-contained and cannot be evaluated because there is no explanation of the formulas actually used in the theoretical analysis or the temperature conditions. A minimum of formulas, a definition of what is being plotted on the graph, and an explanation of the parameters are necessary. In addition, data in a wider parameter range is also necessary to see the robustness of the results.

Version 1:

Reviewer comments:

Reviewer #1

(Remarks to the Author)

I can partially agree that the authors have addressed the comments by both reviewers. But I am still not fully convinced by the reply and would like the authors to address the following two points.

1. The authors say that the FFT of the Fermi surface is not a convincing evidence to determine the periodicity. But I hope they can at least show one figure of the FFT, not just saying that "FFT is not straightforward and ACP is better".

2. The authors say that it is difficult to determine the Pomeranchuk instability temperature from theory due to the complexity of the theory. Then, I hope they can perform temperature dependent ARPES and Fermi surface measurements as well as ACP analyses and show the transition temperature from the experimental point of view. This should give new insights into the correctness of the theory.

Reviewer #2

(Remarks to the Author)

The revised manuscript shows certain improvements in the logical structure compared to the previous version, where logical leaps were frequently observed, and overall, the quality of the paper as an academic article has improved. However, even after these revisions, the novelty of the results still feels somewhat lacking, and considering that the impact of these findings on the broad readers in Nature Communications may not be substantial, I find it difficult to recommend publication.

As both referees have pointed out, a prior ARPES study on the same material (Nat. Phys. 19, 1591 (2023)) has already observed C_2 symmetry. Therefore, the novelty of this work lies rather in the identification of the orbital character associated with this symmetry. However, even within this scope, the only conclusions that can be safely drawn from the present experiments—without speculative inference—are that the α pocket is composed of even orbitals, and the β and γ pockets consist of a mixture of even and odd orbitals. Beyond this, the paper's conclusions are based on multiple layers of inference, and it would be extremely difficult for readers who are not specialists in ARPES, DFT, or fRG to assess how reliable these conclusions truly are.

One example is discussed in [Response to the 1st comments of referee 2], where one layer of inference involves identifying whether orbitals are in-plane or out-of-plane. The authors cite the agreement between ARPES data in Fig. S4(c) and the DFT results in Fig. S4(f), as well as the polarization-dependent behavior (extended), as justification for fully trusting the DFT orbital character. However, it is difficult to accept that the entire orbital dependence of the DFT calculation can be trusted

solely because it reproduces this ARPES signal. Furthermore, the structure of the manuscript does not present a logical, step-by-step narrowing down of orbital characters. As a result, the readers may find a significant gap between the experimental results and the authors' claims, which is hard to bridge. Questions also remain regarding the reliability of both the polarization-dependent ARPES measurements and the DFT calculations themselves [Comments 1 and 2 by referee 1].

Regarding the second main point—the proposal of a Pomeranchuk instability—while this scenario can be understood as one possible explanation for the origin of C_2 symmetry, it is questionable whether it is the only plausible scenario. Even if a correct band structure is used as input, the reliability of the resulting prediction depends on the limitations of the theoretical method, and cannot be guaranteed. In this case, the 1-loop fRG analysis is performed with a step function for temperature, which undermines its quantitative reliability [Comments 3 by referee 1, 10 by referee 2].

Finally, one more point worth noting is found in [Comment 12 by referee 2], where the authors claim that the Pomeranchuk instability can be understood from the bare susceptibility shown in Fig. 3(f) as originating from in-plane orbitals. However, the momentum dependence of the bare susceptibility peaks away from the Γ point, making this interpretation unclear. It would be more appropriate to directly perform an RPA calculation by adding U and V to the bare susceptibility—even using a scaled-down approximation—so that this point can be more convincingly demonstrated.

Version 2:

Reviewer comments:

Reviewer #1

(Remarks to the Author)

I am satisfied with the authors responses and the manuscript can be published in its present form.

Reviewer #3

(Remarks to the Author)

In this manuscript, Chiara Bigi et al. report a systematic investigation of the mechanisms underlying nematicity in CsTi_3Bi_5 . The authors employ polarization-dependent ARPES to reveal the orbital-selective nematic deformation and support their findings with FGR calculations, which indicate a d-wave Pomeranchuk instability driven by orbital-dependent electronic correlations and chemical potential detuning away from VFs. The paper is well written and the explanation of the result is well addressed. However, I have significant concerns on its novelty and broad impact appear limited in the context of recently published works. The implications of these results are not striking enough or new. In this respect, this work may not be suited for general audience targeted by Nature Communications, but for more specific audience who is investigating this material. The followings are my concerns for authors' consideration.

1. The use of polarization-dependent ARPES to elucidate orbital contributions and autocorrelation methods to demonstrate electronic nematicity has been previously established in related kagome metals. Notably, orbital-selective behavior was reported in RbTi_3Bi_5 (Nat. Phys. 19, 1827, 2023), and clear autocorrelation evidence of nematicity was provided in the same compound (Nat. Commun. 14, 4892, 2023). The current manuscript does not appear to significantly advance beyond these earlier works. Especially, the autocorrelation ARPES result (Nat. Commun. 14, 4892 (2023)) is better than that in this manuscript.
2. In the response to the Reviewer#1, the authors claim that "Ti-based kagome samples are highly reactive, and even mild thermal treatment in an inert Argon atmosphere can cause sample degradation". However, previous work on RbTi_3Bi_5 (Supplementary Fig. 14 in Nat. Commun. 14, 4892 (2023)) showed clear C_2 anisotropy persists at 200 K with negligible aging effect. This discrepancy requires clarification. Additionally, as shown in Extended Data Fig. 7, the autocorrelation maps at 16 K, 250 K and RT are much clearer than these at 90 K, 123 K, 200 K and 270 K. This non-monotonic changes in autocorrelation clarity across temperatures are not convincingly explained by aging effect alone.
3. The anisotropy observed in autocorrelation maps could arise from matrix element effects rather than intrinsic electronic ordering. To exclude this possibility, the authors should perform measurements with rotated sample orientations to verify whether the scattering vectors rotate accordingly, which is a standard validation step currently missing in the manuscript.
4. In their response to the Review#2, the authors state that "the only ARPES data available to date [Nat. Phys. 19, 1827 (2023) for RbTi_3Bi_5 not for CsTi_3Bi_5 , Nat Commun. 14, 4089 (2023)] does not report a nematic state and focus only on the topological features of the normal state band structure". This claim is inaccurate. In fact, the ARPES study published in Nat. Commun. 14, 4089 (2023) provides clear evidence of electronic nematicity via autocorrelation analysis (see Figures 5 and S11–S16), with data quality and resolution that appear superior to those presented in this work. Moreover, the authors emphasize that "As a matter of fact our study is the first to report nematicity in CsTi_3Bi_5 Ti-based kagome material from ARPES". Given the strong electronic similarities between CsTi_3Bi_5 and RbTi_3Bi_5 , and considering that robust nematicity has already been established in the latter through ARPES measurements, insisting on being "first" for the Cs compound seems unnecessary and does not substantively strengthen the manuscript's contribution.

We sincerely thank both referees for their constructive feedback and valuable suggestions. Their insights have been instrumental in improving the quality of our work. Below, we provide a point-by-point response, detailing the changes made to the manuscript and addressing all concerns raised.

Reviewer #1 (Remarks to the Author):

The manuscript of Bigi et al shows ARPES results together with theoretical study on the origin of the electronic structure showing nematicity in CsTi₃Bi₅. I think the work has been conducted nicely and the paper is interesting. Basically, it can be said that this is a follow up of what is reported in Nature Physics 19, 1591 (2023) using ARPES in terms of experiment. The theoretical part is the novel part, suggesting a new origin of the observed Pomeranchuk instability. I suggest the authors to consider the following points before I can recommend it for publication in Nat. Comm.

We appreciate that the referee acknowledges the quality of our work and finds the results interesting. The work Nat. Phys. **19** 1591 (2023) cited by the referee has been seminal in this field. While the evidence there pointed to the increasing role of Coulomb interactions, particularly in relation to the multiorbital nature of the system in driving the observed phase, this aspect remained speculative, and no direct proof could be presented. As the referee highlights, the real novelty of our work is the identification of the underlying mechanism as a Pomeranchuk instability. Crucially, uncovering this required precise orbital-resolved measurements, and this is where ARPES played a unique role.

Our ARPES approach was specifically designed for this purpose, employing multiple light polarizations, including horizontal and vertical linear polarization, as well as right- and left-handed circular polarization. This methodology allows for direct orbital probing, leveraging the initial-final state symmetry match. In contrast, STM and QPI techniques lack the necessary selectivity to resolve orbital contributions in a comparable manner.

Another critical aspect is that resolving the electronic instability required detailed knowledge of the Fermi surface contours and the momentum-resolved energy dispersions with explicit orbital selectivity. While STM can provide partial information in this regard, its momentum resolution is far inferior to that of high-resolution, multi-polarization ARPES. Thus, ARPES was instrumental—alongside our DFT analysis—in elucidating the microscopic origin of nematicity in CsTi₃Bi₅, which had remained unresolved even in previous studies.

Moreover, our ARPES measurements provided an important validation of the Wannier functions-based downfolding model, especially for the binding energy of the p_z pocket, that was fitted by comparison to ARPES.

Generally speaking, the full ab-initio model is too demanding to be analyzed within the FRG scheme. In this respect, downfolding is strictly necessary and requires validation by experiment data. Our manuscript provides an instance, where both theory and experiment are working hand in hand to provide a well founded understanding of the nematic transition in a kagome metal.

To further clarify these points, we have now improved the manuscript, making the connection between ARPES and DFT more explicit in the text.

1. The authors refer to the autocorrelation maps (ACMs) of the Fermi surface to demonstrate the nematicity. I think it is important, but I would like the authors to first perform FFT on the raw data of the Fermi surface first before discussing the ACMs; it should be similar to the analysis of Nature Physics 19, 1591 (2023), and although it maybe not so clear, one should be able to recognize the symmetry lowering at this stage. If this is not the case, I cannot really trust the ACMs.

The authors also should add an explanation of the ACMs. How it is derived (the equation), the physical meaning, etc.

We thank the referee for this suggestion. However, in practice, this is not straightforward. The Fourier transform of a Fermi surface obtained from angle-resolved photoemission spectroscopy (ARPES) can, in principle, provide insight into the underlying electronic structure in real space. As the referee correctly points out, it may reveal symmetry breaking, such as nematicity, charge density waves (CDW), or nesting.

However, applying this method requires that the periodic unit being transformed is sampled multiple times. In other words, ARPES would need to probe several repeating Brillouin zones simultaneously, which is not feasible, especially if one aims to maintain the necessary momentum and energy resolutions required for this study. Additionally, even if this were possible, the analysis would be complicated by geometric matrix element effects arising from the rotational symmetries needed to access neighboring Brillouin zones.

This contrasts with scanning tunneling microscopy (STM), which operates in real space. Since STM naturally captures multiple periodic units, its Fourier transform provides information in reciprocal space that can, to some extent, be compared to the ARPES-derived Fermi surface. However, ARPES retains an

advantage in its sensitivity to orbital character due to light polarization, which quasiparticle interference (QPI) does not directly provide. Notable examples of this approach can be found in the works of the M.P. Allan's group, such as Battisti et al. (<https://www.nature.com/articles/s41535-020-00292-4>).

The autocorrelation of an experimental Fermi surface map is the correct method for extracting periodic information, as it can be performed within a single Brillouin zone while maintaining consistent matrix element conditions. This approach relies on a self-convolution operation applied to the ARPES intensity data in momentum space. Mathematically, if $I(k)$ represents the ARPES intensity at momentum k , the autocorrelation function is given by:

$$A(q) = \int I(k)I(k+q) dk$$

Therefore, due to the exact mathematical formulation of this approach, which we have implemented in our manuscript, the autocorrelation method (ACM) enables the automatic identification of scattering vectors. Peaks in $A(q)$ correspond to characteristic wave vectors q that connect different regions of the Fermi surface. These wave vectors often signify quasiparticle interference (QPI), which is also observed in STM experiments. This is precisely why ACM, rather than a Fourier transform (FFT), is the appropriate method for comparison here. An FFT would not be appropriate in this context.

Another key advantage of ACM is its ability to reduce noise by enhancing symmetric features of the Fermi surface while suppressing asymmetric ones. This makes it particularly useful for identifying periodic structures, especially in materials with reconstructed or multi-component Fermi surfaces.

It is important to note that ACM and FFT are however closely related in that both reveal periodic modulations in momentum space. In materials with charge order, nematicity, or unconventional pairing, autocorrelation analysis can uncover periodic modulations that may not be immediately apparent in raw ARPES data.

We have now added a proper description in the methods section to further support our choice.

2. In Figs. 1 d and e, the experimental data is compared to the DFT calculations. The overall consistency is good, but near the Gamma point at -0.6 to -0.8 eV, there are some deviations. The authors should discuss this point.

We thank the referee for noticing this discrepancy. As it has already been reported in the literature, the additional spectral weight in the ARPES data is stemming from a topological surface state (TSS) that is not recovered within DFT surface calculations (see *Nat. Commun.* **14**, 4089, 2023). Consequently, we also do not see this additional band in our DFT surface band structure in the Supplemental Information Figure S1.

Nonetheless, since we are interested in Fermi surface instabilities and hence in the electronic states around the Fermi level E_F , this discrepancy is not crucial for our discussion of the Pomeranchuk instability and we feel encouraged to proceed with our theoretical analysis without considering the TSS.

We have added a discussion of this fact in the Supplemental Information I and thank the referee again for their suggestion.

3. The authors should explain what is the temperature that the Pomeranchuk instability occurs in the experiment, and how can it be explained from the present theory.

This is a very important question. Experimentally, as in previous studies, we performed ARPES at a temperature where the sample is already in the nematic state. However, Ti-based kagome samples are highly reactive, and even mild thermal treatment leads to rapid sample degradation, causing an immediate loss of resolution and, in many cases, complete disappearance of the bands.

As detailed in the Methods section, meticulous sample preparation was essential, including glovebox handling to minimize degradation. This sensitivity makes it challenging to experimentally determine the nematic transition temperature. However, for similar materials, the transition temperature is generally higher. For instance, in V-based kagome systems, it has been reported to be 35 K (see *Nat. Commun.* **14**, 3899, 2023). In another study (*Nat. Commun.* **14**, 4892, 2023), the authors also faced difficulties in extracting this information experimentally, for similar reasons to us. Nevertheless, as in our case, the observed C_2 symmetry serves as clear evidence that the system is already in the nematic state, which is the main focus of our work.

From the theoretical side, it is well known that the FRG within the single loop approximation fails to quantitatively predict the critical temperatures. This insufficiency can be lifted by including multi-loop corrections to the FRG flow equations. However, multi-loop corrections have been so far applied only to the single-band Hubbard model, because their numerical complexity hinders any

application to multi-orbital systems, and are hence non suitable for the treatment of our realistic model for CsTi₃Bi₅.

Still taking the critical scale Λ_C of our FRG flows as a proxy for the critical temperature, we encounter a sensitive dependence of Λ_C on the chosen interaction parameter. Recording the phase diagram in the newly added Supplementary Information Figure S6 we see that the Pomeranchuk instability is robust over a broad range of interaction values and features a critical scale spanning multiple orders of magnitude. Therefore, the proposed mechanism for the nematic transition is compatible with a broad range of possible transition temperatures.

In particular, the transition temperatures expected from theory are larger than the temperature of the ARPES experiments ($\sim 15\text{K}$), clearly supporting the presence of a finite nematic order parameter.

We have added a detailed discussion about the estimation of the critical temperature from the FRG calculations in the Supplemental Information V and thank the referee for making us aware of this missing piece of information in the previous version of the manuscript.

Reviewer #2 (Remarks to the Author):

This paper is composed of two parts. The first half is the experimental results of polarization-dependent ARPES performed on CsTi₃Bi₅. As a result, the authors found that of the four Fermi sheets (α , β , γ , δ), the α and δ have the character of out-of-plane orbitals, and the γ has the character of $d_{xy}, d_{x^2-y^2}$. It was proposed that the β has d_{xz}, d_{yz} character. It was also confirmed that the rotational symmetry breaking phase (C2) proposed for this material occurs on the γ fermi surface. In the second half, which is theoretical, the origin of the C2 state was proposed based on many-body calculation techniques (fRG). As a result, a Pomeranchuk state in which an imbalance in electron density occurs was proposed.

In various Kagome metal systems, it is well known that orbital properties are important to understand a lot of phase transitions. Thus, the present attempt to detect the orbital dependence of the fermi surface using polarization-dependent ARPES can be evaluated to a certain extent as valuable. On the other hand, there are many concerns where the theoretical logic for interpreting the experimental results is unclear or explanations are missing. Throughout the manuscript, the authors' conclusions are often exaggerated and leaps are made in comparison to the experimental results actually obtained. Therefore, this manuscript does not reach the level required for publication in Nature

Communications. I recommend that the gap between them be properly filled and that the paper be considered for publication in more specialized journal.

We agree that orbital character has a crucial role in understanding phase transitions, and this holds true for a wide range of materials. However, our study is not merely an investigation of orbital character; rather, it aims to elucidate how different orbital manifolds contribute to reducing the system's symmetry to C_2 . We apologize if this was not conveyed clearly in the manuscript and for any missing explanations. In addressing the referee's concerns, which we have taken seriously, we have made improvements to the manuscript, ensuring that our arguments and discussions are presented with greater clarity.

That said, we were somewhat surprised by the referee's concern that our conclusions may be exaggerated. After carefully re-evaluating our statements, we believe that we have been both honest and fair in our presentation. Prior to our study, a detailed understanding of the orbital character in this system was lacking. Furthermore, while previous STM-based studies identified nematicity, they could not determine its microscopic origin beyond a speculative level. In contrast, our work establishes this connection by leveraging various light polarizations in ARPES and developing a theoretical framework that suggests a Pomeranchuk instability as a plausible driving mechanism. This represents a clear and substantiated conclusion based on our experimental and theoretical findings.

Once again, we emphasize that our study extends beyond a simple orbital analysis. By placing our results in the broader context of previous studies, we propose a mechanism that provides a coherent and elegant explanation for both past and present experimental observations.

We hope that the referee is pleased by the revised version of the manuscript, where we tried to make the strong bond between theory and experiment more translucent.

The main concerns are as follows.

1. The authors concluded that the following in-plane/out-of plane orbital dependence can be obtained from the Figure 2 (a, c). However, there appears to be jump between their conclusion and the experimental results.

"Our findings reveal that the inner circular pocket (α) centered at Γ is predominantly composed of orbitals with out of plane symmetry. Additionally,

the outermost hexagonal Fermi surface sheet (δ), forming the pockets around both the M and K points, exhibits similar characteristics. In contrast, the two internal hexagonal Fermi surface sheets (β) and (γ), rotated by 30 degrees relative to one another, exhibit a substantial contribution from in-plane orbitals"

This is because, from the linear polarized ARPES experiment, we estimate that the parity with respect to the mirror plane can be identified, then wouldn't the experimental results in Figure 2 (a,c) only reveal the parity with respect to the mirror plane? It seems not self-evident how the authors' conclusion that "We have discovered that the α and δ planes have the character of interplane orbitals, and the β and γ planes have the character of in-plane orbitals " is derived from this. The clear relationship between the light-polarization (LH and LV) and orbital symmetry should be formulated in the main text.

We agree with the referee that our initial description was too simplistic. First, we emphasize that parity is defined with respect to the mirror plane. To better capture this parity, the experiment was performed, as also shown in the Supplementary Information, using multiple sample rotations. Additionally, we repeated the measurements on various crystals, consistently obtaining the same results.

We acknowledge the need for a more detailed discussion, and in response to this concern, we have removed the oversimplified description and provided a significantly more precise explanation in the Supplemental Information IV. Specifically, in a photoemission experiment, the intensity from ARPES strongly depends on a factor called photoemission matrix element: $|M_{fi}(k)|^2$

This term is proportional to $|\langle \phi_f^k | \epsilon \cdot r | \phi_i^k \rangle|^2$. Here ϵ is a unity vector along the polarization of the vector potential A . This means that during the photoemission process, the photoemission intensity will vanish unless that term is different from zero, which strongly depends on the symmetry of the initial state wavefunction and the one of the final one. In general, it is different from zero when that object is an even function with respect to the system's mirror plane. In experiments, at the detector, it is reasonable to consider a spherical symmetry as the final state, so our discussion will have this caveat, which is what generally is assumed. This is also why we consider photoemission with respect to the mirror plane, if our detector is away from the mirror plane there would be the lack of a well-defined odd or even symmetry for it. That said, to have non-vanishing photoemission matrix elements, the initial state at the detector must be such that $\epsilon \cdot r | \phi_i^k \rangle$ is even too. Our problem can be therefore

reduced to the symmetry of $|\phi_i^k\rangle$. Still, referring to the mirror plane containing the incident light, according to the schematics we described in the main text and shown also in the context of ACM, and assuming the detector corresponds to a final state represented by a plane wave of even parity ($|+\rangle$), only transitions from initial states of even parity are allowed - specifically, $|+\rangle$ and $|-\rangle$. In contrast, transitions involving initial states of odd parity are forbidden ($|\mp\rangle$), leading to a vanishing photoemission intensity. The example shown here should be done for all orbitals.

We use the information from the dichroism maps to indirectly probe the in- and out-of-plane character of the Fermi surface sheets:

The linear dichroism can be easily compared with the parity eigenvalue of the Fermi surface states w.r.t. the respective mirror plane. This has been carried out explicitly in a newly added section in the Supplemental Information and shows a very good agreement between ARPES and DFT data (Figure S4). Because this comparison proves the accuracy of the orbital weights redistribution on the Fermi surface in the theoretical model, we can use the DFT data to infer the in- vs out-of-plane character of the Fermi surface sheets that is not directly accessible by ARPES due to the measurement geometry.

We have also highlighted this logical thread in the main text and hope to thereby clarify potential misunderstandings.

2. In relation to the above, even if we consider a d_{xy} orbital on kagome lattice, there are three sublattices. Then, three orbitals do not form 90 degrees, so it seems non-trivial how the symmetry argument corresponds to the results of this study, and a proper explanation is required.

We thank the referee for making us aware of this potential source of misunderstanding.

The Wannier model was constructed employing atomic orbitals (i.e. eigenstates of the angular momentum operator) with a global quantisation axis, i.e. the same for each sublattice site in the kagome unit cell. Hence, if we talk about the d_{xy} orbital in the manuscript, this is really the real spherical harmonic with quantum numbers $l=2, m=-2$.

We have explicated this in the Methods section of the manuscript and also provided real space images of all orbitals in the Supplementary Information Figure S3.

An exemplary arrangement of d_{xy} and $d_{x^2-y^2}$ orbitals on the lattice is further provided in Supplementary Information Figure S4, where also the relative orientation on the lattice becomes apparent.

3. Furthermore, the authors conclude that the γ plane is d_{xy} , $d_{x^2-y^2}$ and the β plane is d_{xz} , yz , based on the results of changing the slit direction from the original Γ -M to the Γ -K direction, but it is difficult to understand because they do not quote the Figure numbers in the text. The actual experimental results are presumed to be extended data Fig.5, but shouldn't this Figure be included since it is important for identifying the orbit? In addition, a discussion of the symmetry argument to derive these results should be included using equations, etc.

" we also performed light-polarization-dependent ARPES measurements across various experimental geometries, including rotations of the analyzer slit along the Γ -K direction. The response of the two hexagonal Fermi surface sheets to this rotation is notably distinct."

The referee is correct, and we believe that this has been now covered by a more general argument in the text. One thing we urge to stress is that such a precise attribution does not come from ARPES alone. It comes from orbitally resolved DFT calculations (new Supplementary Information Fig. S3 and S4), which agrees with orbitals having parity as the one elucidated by ARPES. We totally realized that in our previous discussion, this was too simplistic and was not correct. We have now replaced it with a more general statement which is based on symmetries.

4. In identifying the orbitals of the β and γ planes, it is necessary to clarify the starting point for the orbital candidate trajectory and how the trajectory was narrowed down based on the each h experimental results (with appropriate reference to the Figure). For example, why do p orbitals, s, $d_{3z^2-r^2}$ were excluded?

Since we are interested in the bandstructure around the Fermi level, we only consider atomic orbitals from the valence shell of the respective atoms. Consequently, as stated in the method section, we include Cs 6s, the full 3d shell of Ti and 6s and the full 5p shell of Bi in our calculations and calculate the Fermi surface weights for all of these orbitals. The single-particle spectrum of

the wannierised model matches the DFT bandstructure perfectly within a window of 4eV around the Fermi level justifying the choice of atomic orbitals to construct the Wannier model.

To reveal the parity character of the eigenstates we have calculated the Fermi surface weights for all of these orbitals to reveal the parity of the Fermi surface states and compare them with the ARPES data. It turns out, that the s orbitals do not contribute to the Fermi surface at all. For all other orbitals, we display the Fermi momentum resolved orbital weights in the new Supplemental Information Figure S3 and also added an additional note in the method section on how we calculate the orbital content of the Fermi surface. From this, it becomes directly evident that also the dz^2 orbital does not contribute to the orbital makeup of the Fermi surface.

5. From the results, shouldn't they include information on which orbitals are more prevalent at which wave number k on which Fermi surface? They concluded that there are two types of orbitals, d_{xy} and $d_{x^2-y^2}$ for the γ and d_{xz} and yz for the β , but are the orbital weight the same along k , or do they change? Including this result would also be helpful for comparison with first-principles calculations."

We have added an orbital resolved Fermi Surface for each orbital in the Supplemental Information Figure S3 to show the individual contribution to each pocket. Since the orbitals have a global quantisation axis and do not rotate with respect to the different kagome sites, their contribution shows a strong anisotropy also detectable by the linear dichroism data of ARPES provided in Supplemental Information Figure S4.

However, also the Fermi surface maps of Figure 3a,b in the main text are momentum resolved Fermi Surface weights. But via the summation of both all in-plane (blue color) and all out-of-plane (red color) orbitals, this anisotropy is obscured and the Fermi surface weights appear isotropic along the Fermi surface pockets.

We have added an additional note in the Methods section on how we arrive at Figure 3a,b and thank the referee for clarifying this inaccuracy.

6. Between extended date Fig. 5(d) and Fig. 2(d), the asymmetry of C2 appears to be more clearly suppressed in Fig. 5(d). What is the reason for this?

We do not know why this happens, but even if more suppressed it is still present. It is important to mention that part of the functional shape of the lines might be also due to matrix elements. Indeed, matrix elements were different already in the Fermi surface maps, and this could be part of the reason. Importantly, for all samples used in the experiments also other cleaves and other samples data collected during the experimental time and all rotation the asymmetry is still present, and all the time C_2 was observed in the autocorrelation maps.

7. The definition of the orbital basis should be explained. The authors call the orbitals in this study dxz , dyz , etc., but most readers who are not familiar with this material will not understand which orbital is which in real space. The definition of the orbital should be illustrated.

As already stated in the answer to the referee's concern number 5, we have added explicit plots of the real space orbitals on the lattice in the supplement and explain the orbital basis in the Methods section. We hope this also clears the referee's questions expressed in this concern.

Regarding the theory part, the conclusion is that a Pomeranchuk instability occurs based on a certain model parameter (initial value of the interaction U, V_1, V_2 and temperature). However, since there is almost no information on the robustness of the results, it is impossible to judge how reasonable the results are from the manuscript as it is, and doubts from many readers cannot be dispelled. It should be fundamentally revised.

We would like to stress that we are not treating a model system but simulating an actual material. Consequently, we have chosen interaction parameters in compliance with ab-initio cRPA values for the same material class. These values should represent a realistic starting point for our calculations.

However, we agree with the referee that the ab-initio values for the interaction are not guaranteed to be consistent with the screened interaction profile present in the sample. Since a direct experimental verification of the cRPA values is not possible (unlike a direct comparison of the DFT band structure with ARPES), we have performed a detailed stability analysis by varying independently both U and the decay of the long range tail of the repulsive interactions. Our results summarized in a new paragraph in the Supplemental Information show that the

Pomeranchuk instability is the only abundant phase in the reliable part of the phase diagram. We hope this ensures the referee of the soundness of our theoretical analysis.

8. The authors conclude that such Pomeranchuk instability appear due to the inclusion of many-body effects by fRG compared to the mean-field method, but this conclusion is not immediately convincing because the results of solving the same model at the mean-field approximation level are not presented. The authors should present the mean-field results of solving the same model at the same temperature as fRG with the same magnitude of interactions (Of course, they should verify the robustness by changing U, V_1, V_2 .). In particular, can they say that the Pomeranchuk instability (PI) does not appear in the mean-field approximation even if the magnitudes of the long-range Coulomb interactions V_1 and V_2 become somewhat large? If PI appears within the mean-field model, the validity of the magnitude of the interactions V_1, V_2 in this study must be explained.

The Pomeranchuk instability (PI) should not be seen as “a beyond mean-field (MF) effect”. In the manuscript, we explicitly claim the opposite: even when taking into account quantum fluctuations within a proper many-body analysis by means of the FRG, the PI already appears on the MF (or equivalently RPA) level due to significant long range interactions. To make this more clear to the reader, we now show it explicitly by calculating FRG flows with the same parameters in the Supplemental Information V but without considering cross-channel contributions between the different diagrammatic channels present in the FRG flow equations of Figure 4a. This is equivalent to an RPA (i.e. MF) resummation of the different channels. In the charge channel, we likewise find a dominant PI (when compared to a CDW transition). We hope that these additional calculations added to the Supplemental Information are suitable to clarify potential misunderstandings.

In addition, we would like to stress two important points of the theoretical analysis:

- 1) Firstly, the FRG is strictly better than RPA (and hence MF), since it comprises more diagrams in the expansion of the partition function. Hence, it is always justified to perform FRG calculations to check for the stability of MF/RPA results upon considering quantum fluctuations.
- 2) Secondly, especially in systems with many competing instabilities like the kagome compounds, a careful examination of all competing orders as

well as their competition and cooperation (both absent in MF/RPA) is mandatory for reliable predictions. Hence, any MF theory has to be justified by an actual many-body calculation. Our study here shows a direct example of the importance of this procedure: In PRB **111** 125153 a combination of FLEX and density wave equation has reported a nematic odd-parity bond order in CsTi₃Bi₅. Comparing their result with our present FRG analysis, which contains more diagrams, we can attribute their result to a bias in evaluating the cross-channel feedback introduced by the density wave equation.

Therefore, the fact that FRG and MF/RPA give the same results should not be taken as granted but rather as an important result of our study.

9. It is known that low-temperature calculations are difficult in such mesh fRG because it is generally difficult to obtain fine k-meshes. However, was this calculation performed at a temperature that is consistent with the transition temperature of about 100K observed in experiments? If the answer is Yes, it should be discussed whether this is a reliable region with the present number of k-mesh. If the answer is No, is there a guarantee that the same PI will be dominant at a low temperature of around 100K? These temperature dependence needs to be discussed.

We thank the referee for making us aware of the missing information in the Methods part of the manuscript.

The FRG calculation was performed with a sharp frequency cutoff, i.e. all flows are at zero temperature. However, the constraint region of Matsubara frequency axis induces a broadening of the fermionic loop integrals comparable with a temperature. Hence, we can use the flow scale Λ as a proxy for the effective temperature of the system even though such a relation should be treated with care.

We have estimated the energy resolution by calculating the density of states, obtaining converged results up to energy discretizations of 1 meV. This is well beyond the critical energy scales with the given parameters (and also for almost all flows in the phase diagram of Supplementary Information Figure S6), where PI is the dominant instability. The only phase space region, where we can no longer trust the FRG results is when a magnetic order is surfacing, since there the critical scales are comparable to our energy resolution and numerical artifacts are expected.

We have added a detailed discussion about the estimation of the critical temperature from the FRG in the Supplemental Information V and hope the referee is satisfied by our delineations.

Experimentally, however, we have never observed a transition in this system at 100 K. In other compounds such as CsV₃Sb₅, there is a CDW which appears at 98 K which has been identified as the cause of nematicity. However, the CsTi₃Bi₅ system is nematic without CDW, as is well-known from many previous studies. In addition, as in those previous studies, we have performed our ARPES measurements at a temperature where the sample is already in the nematic state. However, Ti-based kagome samples are highly reactive, and even mild thermal treatment leads to rapid sample degradation, causing an immediate loss of resolution and, in many cases, complete disappearance of the bands.

As detailed in the Methods section, meticulous sample preparation was essential, including glovebox handling to minimize degradation. This sensitivity makes it challenging to experimentally determine the nematic transition temperature. However, for similar materials, the transition temperature is generally high. For instance, in V-based kagome systems, it has been reported to be 35 K (see *Nat. Commun.* **14**, 3899, 2023). In another study (*Nat. Commun.* **14**, 4892, 2023), the authors also faced difficulties in extracting this information experimentally, for similar reasons to us. Nevertheless, as in our case, the observed C₂ symmetry serves as clear evidence that the system is already in the nematic state, which is the focus of our work.

With the lack of an experimental estimate for the nematic transition temperature, from a theory perspective, the newly added FRG phase diagram in the Supplemental Information Figure S6 shows that the prediction of a PI prevails for critical scales over several orders of magnitude, depending on the chosen interaction parameter. In particular, the predicted critical scales are larger than the experimental temperature of 15K, for which a clear finite nematic order parameter is present.

Therefore, even though we can not predict a precise value for the transition temperature both by experimental and theoretical means, our many-body results are consistent with the experimental realisation of a nematic state in CsTi₃Bi₅ at low temperature irrespective of the details of the interaction Hamiltonian.

10. In Fig. 4(b), the divergent growth of two-particle interaction vertex does not stop as a function of $\Lambda \sim 10000\text{K}$. On the other hand, what would happen if they used parameters U, V that can be renormalized up to around $\Lambda = 100\text{K}$. Such parameter region would be more consistent with the experiment."

The FRG flow displays a divergence upon encountering a phase transition. Hence, we cannot continue the flow across the transition and therefore terminate it upon a divergence that marks the breakdown of the one-loop truncation. As a consequence the size of the order parameter within the symmetry broken phase can not be estimated via the FRG. A direct comparison of the critical scales of the FRG and the nematic transition temperature is hampered by several fundamental hurdles:

- 1) First, as already stated in the reply of the previous concern, we perform a sharp frequency flow rather than a temperature flow. Hence the scales in the FRG can not be directly associated with a temperature.
- 2) Second, it is well known that the FRG within the single loop approximation fails to quantitatively predict the critical temperature, consistently overestimating it. This insufficiency can be lifted by including multi-loop corrections to the FRG flow equations. Their numerical complexity prevents any application to multi-orbital systems and they are hence non suitable for the treatment of CsTi₃Bi₅.

However, we still see in the phase diagram of Supplemental Information Figure S6 that such critical scales of 100K can be reached within our FRG flows for suitably small interaction parameters and the PI still prevails. We want to point out that such parameters are way smaller than the cRPA prediction for realistic kagome metals.

Additionally, a transition temperature considerably higher than 100K cannot be ruled out by experimental means: As stated in the reply to the previous concern, the experimental determination of the nematic transition temperature has not been possible thus far neither by our experiment nor other ones. The only secure experimental evidence is the presence of a nematic state at temperatures at which the experiments are performed.

With the lack of experimental accessibility of the nematic transition, we think the first sound theoretical analysis of this transition is highly valuable for the community even more as it appears to be very robust with respect to the chosen interaction parameter and thereby the obtainable transition temperatures.

11. In relation to the above, what is the estimate of the C2 transition temperature based on the fRG theory?

We hope to have answered this question already by dealing with the previous concerns of the referee and the changes applied to the manuscript in order to resolve them.

12. I Since the main text only provides results for non-interacting susceptibility in Fig.3 (f), in order to compare with the fRG results, the results for susceptibility analyzed at the mean-field level using the same model as fRG ($J = 0.8$ eV, $U = 4$, $V1 = 1.5$, $V2 = 0.5$) should be presented and compared.

We would like to emphasize that we present the bare susceptibility in Figure 3f to develop a physical intuition on the dominant fluctuations of the system. In particular, we conjecture from the bare susceptibility that the in-plane orbitals dominate the low energy excitations of the system.

Guided by this knowledge, we choose the in-plane d-orbitals (d_{xy} , $d_{x^2-y^2}$) as interacting subspace in our FRG analysis, since all other orbitals are seemingly not contributing to the system's ordering propensities. This is a posteriori justified by comparing FRG calculations with and without the other d-orbitals included in the interacting subspace, which shows consistent results: The main nematic symmetry breaking is encountered on the in-plane orbitals.

The RPA susceptibility cannot be computed for the parameter set suggested by the referee due to its small convergence radius: An interaction scale of $U \sim 3$ eV is triggering a Stoner transition to an ordered phase since screening effects present in the FRG are missing in the diagrammatic ladder resummation. Therefore, we do not show the RPA susceptibility to compare with our FRG results but present the RPA/MF phase diagram in the Supplemental Information Figure S6 alongside with the FRG results.

13. The validity of the results cannot be judged because there is no definition of PI, CDW, SDW, and SC in Fig. 4(b). The definition should be written in terms of wave number dependence and orbital dependence. Presumably, using the center of mass momentum Q of the particle-hole pair, the PI is $Q = 0$ CDW and the CDW is $Q \neq 0$ CDW. In addition, when writing Fig. 4(b), is a specific wave number $Q^* \neq 0$ fixed? In that case, has the possibility of the CDW for other Q being larger been eliminated? Also, does this fRG formalism take into account the bond-type CDW proposed in the Vanadium-based kagome systems? The k -dependence of the obtained vertex should be explained. Also, where did the spin channel for $Q = 0$ disappear to?

We thank the referee for making us aware of the hard accessibility of the FRG data provided in Figure 4. We have added an additional paragraph on the analysis of FRG flows in the Methods part that can be summarized as follows: we do not restrict the exact functional shape of the orders described in the flow diagram of Figure 4b but consider the largest eigenvalue for a given transfer momentum throughout the flow. Hence, the bond-type CDW fluctuations are included in the CDW flow line. While we keep all possible transfer momenta for all three channels (superconducting, charge and spin), we have decided to plot only the four most prominent instabilities in Figure 4b. As the referee has already suspected, the spin channel at $Q=0$ does not feature a sizable eigenvalue at the end of the flow and is hence not shown.

14. The definitions of the terms in-plane and out-of-plane in the color bar in Figure 3(a) are not clear, making it difficult to judge whether the theoretical model is consistent with the experimental results. Which orbital is in-plane dx or dx^2-y^2 ? out-of-plane dxz or yz ? It should be specifically written. Also, does the orbital's character differ depending on the k ? At the same time, it should be shown whether their k -dependence is consistent with ARPES data. Based on that, does the even/odd nature of this orbital match the experimental results?

We thank the referee for making us aware of this imprecise phrasing in the main text. We have now added an explicit definition of the atomic orbitals as already stated in the reply to the previous concerns of the referee and explicitly stated how we derive the orbital weights on the Fermi surface in the Methods part. As the referee has rightly anticipated, the orbital character on the Fermi surface has a momentum dependence that is explicitly shown in the new Supplementary Information Figure S3 and already incorporated in Figure 3a,b of the main text. This has already been discussed previously in this reply.

Moreover, we have now added a direct comparison of the parity of the Fermi surface states with the linear dichroism data in Supplementary Information Figure S4 alongside a discussion about the theoretical background of the comparison.

15. For the theoretical analysis, they should compare the phase diagram by RPA and fRG by changing the values U, V_1, V_2 for showing the robustness of their main result. It is expected that PI disappear when V_1, V_2 are zero or quite small even if they performed fRG. What is the critical value of V_1 and V_2 ?

We thank the referee for his suggestion. In the Supplemental Information, Figure S6 now directly compares the phase diagram from FRG and RPA. As already stated in reply to the referee's concerns Nr. 8,12 the FRG is strictly better than the RPA, since it includes more terms in the diagrammatic expansion of the effective action. Hence, the direct comparison only shows the validity of an RPA calculation and not vice versa. Indeed it seems like the additional cross-channel processes included in the FRG only act as a driver for the critical scale compared to the RPA/MF solution, i.e. increasing the critical scales.

As the referee has correctly anticipated, when the interactions are lowered beyond a critical value, the Stoner criterion is no longer satisfied and we do not encounter a phase transition within our given resolution as can be read off from Supplementary Information Figure S6. Close to the critical value, the FRG results are not reliable anymore and hence the tendencies towards spin density waves can be discarded as unphysical.

16. Shouldn't the results of changing the filling also be discussed using the fRG phase diagram? Since they conclude that the distance of the van Hove singularity (vHS) from the Fermi surface is important for the emergence of the PI, it seems crucial to demonstrate that when the vHS gets closer to the Fermi surface, a different instability switches on."

We agree with the referee that a presentation of the results at van Hove filling was missing in the current manuscript. Since a change of the filling while keeping the bare interaction vertex only influences the bare electronic response, i.e. the kernel of any many-body calculation, we discuss the changed physics in the van Hove filled scenario on the level of the bare susceptibility.

We have calculated the bare susceptibility and diagonalized it in Supplementary Information Figure S5. As expected, the almost perfectly nested hexagonal Fermi surface pocket, again stemming from the in-plane Ti d-orbitals, results in a large contribution close to the M point not present before.

Evaluating the eigenspectrum of particle-hole fluctuations, the peak at the M point corresponding to a 2x2 charge order is comparable with the peak at the Γ point corresponding to a PI transition. Inspecting the associated eigenvectors reveals the peak at M as a precursor for the familiar CBO state on the kagome lattice. The absence of this peak in the pristine scenario (i.e., no filling change)

explains the predominance of PI in the phase diagram as well as the accuracy of the MF/RPA calculations since competing fluctuations are missing.

We thereby conclude that the detuning of the chemical potential is indeed pivotal for the dominance of PI and absence of translational breaking charge order in CsTi₃Bi₅.

Anyway, the paper is not self-contained and cannot be evaluated because there is no explanation of the formulas actually used in the theoretical analysis or the temperature conditions. A minimum of formulas, a definition of what is being plotted on the graph, and an explanation of the parameters are necessary. In addition, data in a wider parameter range is also necessary to see the robustness of the results.

Where applicable, we have given the explicit formulas, while for complex theoretical frameworks such as DFT and FRG we have referred to the documentation of the employed software package and have additionally given the explicit input parameters to enable the reader to perform the calculations themselves. In the analysis of the FRG flows, we have added an additional paragraph where we have sketched the analysing procedure and outlined how to obtain the flow eigenvalues presented in Figure 4b of the main text. In combination with the parameter scans added to the Supplemental Information, an instability analysis at van Hove filling and additional RPA phase diagram, we are confident to have addressed all concerns of the referee within the scope of the present work.

Reviewer #1 (Remarks to the Author):

I can partially agree that the authors have addressed the comments by both reviewers. But I am still not fully convinced by the reply and would like the authors to address the following two points.

We are pleased to hear that the referee is at least partially satisfied with our reply to their concern and thank the referee for their repeated effort in reviewing our manuscript. We hope to also address the remaining issues in this present reply.

1. The authors say that the FFT of the Fermi surface is not a convincing evidence to determine the periodicity. But I hope they can at least shown one figure of the FFT, not just saying that "FFT is not straightforward and ACP is better".

We agree with the referee that a quantitative analysis demonstrating why the FFT of the Fermi surface is not useful is required. From the theoretical side, the FFT requires that several repetitions of the same periodic unit be Fourier-transformed in order to gain meaningful insight into symmetry breaking in the electronic band structure, such as nematicity. This is the case with STM technique, which probes the electronic wavefunction over hundreds of unit cells in the real space (a typical STM image is scanned over $\sim 100 \text{ nm}^2$ area). Thus, ARPES should ideally sample the Fermi surface over hundreds of Brillouin zones (the Brillouin zone being the unit cell of the reciprocal space) to enable the FFT approach to be applied to an ARPES data set with adequate rigour and in full analogy to STM studies. This is inherently prevented by the experimental setup of ARPES, as the technique is limited by the 30° acceptance angle of the detector on one axis and by the range covered by the manipulator's rotation on the other. Other more subtle constraints such as matrix element effects across neighbouring zones and photon energy limitations would further hinder extended-zone sampling. As a matter of fact, the momentum and energy resolutions limit us to probing solely one full Brillouin zone. Consequently, the FFT is not the appropriate approach in the context of this ARPES investigation and FFT treatment of the ARPES Fermi surface will not provide insights into the

real space periodicity of the electronic wave function, as clearly shown by $\text{Re}(\text{FFT})$ reported in Figure 1 below.

Figure 1: Real part of the FFT obtained from the Fermi contour of unpolarised light (CR+CL). Arrows highlight the resulting signal.

We would like to kindly emphasise once again that the autocorrelation method extracts periodic features directly from a **single Brillouin zone** in momentum space. This eliminates the need for data acquisition over **hundreds of Brillouin zones** and provides a more suitable tool for characterising symmetry-lowering phenomena in correlated electron systems through ARPES, as thoroughly discussed in the method section of the main text.

We heartily hope that the referee is now happy with our analysis and agrees with us about the usefulness of the autocorrelation maps.

2. The authors say that it is difficult to determine the Pomeranchuk instability temperature from theory due to the complexity of the theory. Then, I hope they can perform temperature dependent ARPES and Fermi surface measurements as well as ACP analyses and show the transition temperature from the experimental point of view. This should

give new insights into the correctness of the theory.

We thank the referee again for their comment. First of all, we would like to kindly stress that from a theoretical point of view the novelty of our results lies in the microscopic nature of the phase transition, that we reveal as a Pomeranchuk instability, as well as the nematic order parameter. In this regard it is worth noting that our work is only the second work aiming at a theory of nematicity in CsTi₃Bi₅. However, compared to [Phys. Rev. B **111**, 125153] our more advanced FRG calculations reveal a different order parameter, namely an onsite charge imbalance on the in-plane d-orbitals. We attribute this to the bias in the diagrammatic resummation of the previous study.

Therefore, we are interested in the precise shape of the order parameter rather than the critical scale. As the referee has correctly pointed out, it is hard to determine the transition temperature quantitatively.

In the last resubmission, we have provided a large scan in the interaction parameters (cf. Supplementary Information Fig. S6). There we show that the microscopic order parameter does not change over a large range of interaction parameters resulting in critical scales spanning 4 orders of magnitude. This encourages us to trust the order parameter obtained from the FRG calculations irrespective of the exact value of the transition temperature predicted by the FRG since only quantitative changes are expected from multi-loop corrections.

The referee's request of performing temperature dependent ARPES measurements is incredibly challenging. In fact, Ti-based kagome samples are highly reactive, and even mild thermal treatment in an inert Argon atmosphere can cause sample degradation, resulting in a loss of resolution and, in many cases, the complete disappearance of the bands. Furthermore, increases in pressure as well as pressure spikes are common during temperature investigation campaigns. Both are detrimental to the sample, whose signal would consequently fade rapidly.

Nonetheless, we agree with the referee about the importance of a temperature dependent analysis. To overcome the aforementioned difficulties, we first performed a degassing of the manipulator to remove most of the molecules cold-trapped to it, then we gently increased the temperature with a 0.4 K/min ramp to keep the pressure in the chamber $<1 \cdot 10^{-9}$ mbar. Samples were cleaved and measured only after pressure recovered $<2 \cdot 10^{-10}$ mbar. We measured Fermi surface maps with both

CR and CL polarisations to obtain the following autocorrelation maps (ACM) of “unpolarised light” (CR+CL).

Figure 2: Autocorrelation maps obtained from the Fermi surface contours of unpolarised light (CR+CL) probed at the Brillouin zone centre with $h\nu = 65$ eV and for several temperatures. The azimuthal profiles extracted for the Γ -K and Γ -M high symmetry directions clearly show the persistence of the reduced C_2 symmetry of the nematic phase up to room temperature.

We probed several cleaves at temperatures between 16 K up to room temperature (beyond which measurements are technically not feasible), i.e. 90 K, 123 K, 172 K, 200 K, 250 K and 270 K and RT as reported in Figure 2, which we also included in the manuscript as Extended Data Fig. 7 to increase the depth of our study. In all cases, the ACM shows the reduced C_2 symmetry typical of the nematic phase, as highlighted by the azimuthal profile extracted for Γ -K and Γ -M directions where one profile clearly deviates from the other two which instead remain equivalent to each other. Here we want to stress the fact that the variations in broadness and in the profile shapes for different temperature points (e.g. for the 200 K set) are ascribable to the sample aging rather than the onset of a transition. In fact, due to the challenges discussed above, the sample surface quickly aged during the

measurements. Figure 3 shows the ACM and the profiles' set for two temperature points, as measured on both freshly cleaved and aged surfaces. C_2 symmetry persists in all the measurements. However, increased broadening is evident in the aged sample. This renders the C_2 symmetry less apparent, to the extent that it could be mistakenly interpreted for the beginning of a transition.

Figure 3: Autocorrelation maps and azimuthal profiles extracted for the Γ -K and Γ -M directions for a fresh sample surface and an aged one at two different temperatures.

In view of the above analysis, we conclude that $CsTi_3Bi_5$ remains in the nematic phase up to room temperature as no sign of transition was detected. In other words, the nematic transition happens above room temperature, corroborating the evidence of a large critical scale obtained from our theoretical calculations.

We would like to thank the referee once again for stimulating us to perform this important analysis which has significantly improved the quality and depth of our work. We sincerely hope that the referee can now recommend in favour of publication.

Reviewer #2 (Remarks to the Author):

The revised manuscript shows certain improvements in the logical structure compared to the previous version, where logical leaps were frequently observed, and overall, the quality of the paper as an academic article has improved. However, even after these revisions, the novelty of the results still feels somewhat lacking, and considering that the impact of these findings on the broad readers in Nature Communications. may not be substantial, I find it difficult to recommend publication.

We thank the referee for their repeated effort in reviewing our manuscript. However, we strongly have to disagree with the referee's concerns on several levels as detailed below.

As both referees have pointed out, a prior ARPES study on the same material (*Nat. Phys.* **19**, 1591 (2023)) has already observed C_2 symmetry. Therefore, the novelty of this work lies rather in the identification of the orbital character associated with this symmetry.

We would like to rectify the referee's statement regarding the status quo in the field of Ti-based kagome metals:

Upon submission of the manuscript, the C_2 symmetry breaking has only been observed in quasi-particle interference measurements from STM [*Nat. Phys.* **19**, 1591–1598 (2023), *Nat. Comm.* **15**, 9626 (2024)].

The only ARPES data available to date [*Nat. Phys.* **19**, 1827 (2023) for RbTi₃Bi₅ not for CsTi₃Bi₅, *Nat Commun* **14**, 4089 (2023)] does not report a nematic state and focus only on the topological features of the normal state bandstructure. They only speculate on the origin of nematicity reported in the aforementioned STM studies given a more precise knowledge of the normal state Fermi surface and band structure from ARPES. In our work we provide a direct observation of the Fermi surface reconstruction in the nematic state.

We would like the referee to finally realize (after stressing this already several times in the first round of our revision), that the article [*Nat. Phys.* **19**, 1591 (2023)] is an STM work NOT an ARPES study.

As a matter of fact our study is the first to report nematicity in CsTi₃Bi₅ Ti-based kagome material from ARPES.

However, even within this scope, the only conclusions that can be safely drawn from the present experiments—without speculative inference—are that the α pocket is composed of even orbitals, and the β and γ pockets consist of a mixture of even and odd orbitals. Beyond this, the paper's conclusions are based on multiple layers of inference, and it would be extremely difficult for readers who are not specialists in ARPES, DFT, or fRG to assess how reliable these conclusions truly are.

We agree with the referee that accessing the microscopic nature of any many body ground state always includes several layers of inference, since the many body wave functions can not be observed directly in experiment. This is also the case in the previously mentioned STM and ARPES studies and we don't see more substantial obstacles in understanding our work than in any other work combining theory and experiment: From ARPES the two main observables are given by the polarization dependent band structure and the Fermi surface maps. In the article we have compared the DFT band structure and the derived linear dichroism maps with the ARPES measurement finding good agreement. Additionally, we compare the Fermi surface reconstruction in the nematic state predicted by FRG and find good agreement with the AC maps of the ARPES.

We therefore wonder where the referee sees potential to erase layers of inference in our work. For the two main theoretical predictions in our paper (orbital structure of the Fermi surface and microscopic nematic order parameter), we have calculated the accessible experimental observables and they show excellent agreement.

One example is discussed in [Response to the 1st comments of referee 2], where one layer of inference involves identifying whether orbitals are in-plane or out-of-plane. The authors cite the agreement between ARPES data in Fig. S4(c) and the DFT results in Fig. S4(f), as well as the polarization-dependent behavior (extended), as justification for fully trusting the DFT orbital character. However, it is difficult to accept that the entire orbital dependence of the DFT calculation can be trusted solely because it reproduces this ARPES signal. Furthermore, the structure of the manuscript does not present a logical, step-by-step

narrowing down of orbital characters. As a result, the readers may find a significant gap between the experimental results and the authors' claims, which is hard to bridge. Questions also remain regarding the reliability of both the polarization-dependent ARPES measurements and the DFT calculations themselves [Comments 1 and 2 by referee 1].

We would like to stress the dichroism in ARPES is - to the best of our knowledge - the only possible way to access the orbital content of the electronic band structure in experiment. Therefore, the dichroism maps are the only possible benchmark for the accuracy of our DFT model and can be readily calculated without any additional assumptions from the DFT bandstructure. Since dichroism maps obtained from DFT and ARPES data show excellent agreement we have no reason to question the correctness of our results.

We would like to stress that the aforementioned STM works on nematic instability do not have any information about the orbital content of the Fermi surface and hence can't access this information. They rely for their claim of orbital selectivity fully on DFT calculations without the possibility to benchmark the orbital character with experiment.

The comments by referee 1 have been satisfactorily answered already in the last revision. We therefore don't see any need to question the quality of both the ARPES and DFT calculations, which were both performed thoroughly and with state-of-the-art techniques.

Regarding the second main point—the proposal of a Pomeranchuk instability—while this scenario can be understood as one possible explanation for the origin of C_2 symmetry, it is questionable whether it is the only plausible scenario. Even if a correct band structure is used as input, the reliability of the resulting prediction depends on the limitations of the theoretical method, and cannot be guaranteed. In this case, the 1-loop fRG analysis is performed with a step function for temperature, which undermines its quantitative reliability [Comments 3 by referee 1, 10 by referee 2].

We agree with the referee, that several possible microscopic realisations of the nematic states are actively discussed in the community and we have commented on them in our manuscript. And obviously the outcome

might depend on the numerical approach used to solve the quantum many-body problem.

As detailed in the Supplementary Information, we have employed the FRG, since it has proven to be a reliable tool (and even more an applicable tool) to handle electronic instabilities in the weak to intermediate coupling regime for large ab-initio models (see references in the Supplementary Information).

In the case of Ti based kagome metals, only a single theoretical work has thus far been proposed, that employs a combination of FLEX and density wave equation [Phys. Rev. B **111**, 125153]. This procedure selects a subset of the screening processes contained in the present FRG study and hence includes a bias towards exotic bond orders. Our unbiased FRG calculations show that the proposed nematic bond order is not encountered in a more rigorous calculation. Instead the microscopic ground state in the nematic phase is predicted to be an intra unit cell charge density wave on the in-plane orbitals. We further see that this prediction is compatible with both our ARPES measurements and previous STM studies [*Nat. Phys.* **19**, 1591–1598 (2023), *Nat. Comm.* **15**, 9626 (2024)], that did not provide any microscopic explanation for their observations.

We also invite the referee to take a careful read through our manuscript. There is no step function for temperature in our FRG calculations and we can only guess what the referee is referring to. We use a sharp frequency cutoff, not a temperature cutoff for the RG flow. However, it has been shown that the precise choice of the cutoff is not crucial for the predicted leading instability and might only shift critical scales. The same holds true for the multi-loop extensions to the FRG (cf. Paragraph in the Supplementary Information): The 1-loop truncation impedes a quantitative estimate of the critical temperature, but the symmetry breaking and order parameter are reliably predicted.

We would like to kindly note that Comments 3 from Referee 1 and 10 from Referee 2 were already addressed in full detail in our previous revision. We respectfully invite the referee to consider our earlier responses, as the concerns raised appear to reiterate points that we have already clarified.

Finally, one more point worth noting is found in [Comment 12 by referee

2], where the authors claim that the Pomeranchuk instability can be understood from the bare susceptibility shown in Fig. 3(f) as originating from in-plane orbitals. However, the momentum dependence of the bare susceptibility peaks away from the Γ point, making this interpretation unclear. It would be more appropriate to directly perform an RPA calculation by adding U and V to the bare susceptibility—even using a scaled-down approximation—so that this point can be more convincingly demonstrated.

We have already readily replied to the referee's Comment 12 in the previous resubmission. Quoting: "We would like to emphasize that we present the bare susceptibility in Figure 3f to develop a physical intuition on the dominant fluctuations of the system. In particular, we conjecture from the bare susceptibility that the in-plane orbitals dominate the low energy excitations of the system." This means that we've never claimed that the PI can be understood from the bare susceptibility.

Probably the referee is confusing bare susceptibility and MF/RPA. Since the PI is an interaction driven effect (as stated in the manuscript), the bare susceptibility (which does not contain any interactions but only kinetic fluctuations) is not expected to feature a pronounced nematic tendency.

In the previous revision we undertook the effort to calculate the full RPA phase diagram (cf. Supplementary Information Fig. S6) revealing a dominant PI also on the RPA level. In this revision, following the referee's request, we have added an exemplary interacting RPA susceptibility to Fig. S6 (see also figure below for convenience). It features a pronounced peak at the zone center corresponding to nematic fluctuations close to criticality. Notably, this peak develops at scales far above the Fermi level pointing towards an interaction rather than a kinetic fluctuation driven nature of the Pomeranchuk instability. We hope the referee is now pleased by the presentation of our results and trusts the statements put forward in our paper.

We have carefully examined the manuscript and present a more balanced discussion of the closely related works highlighted by Reviewer #3 in the Discussion section. Moreover, we have checked for remaining strong theory claims in light of Reviewer #2's concerns. In this respect, we have changed the phrasing in the Discussion section to highlight that we have performed FRG calculations and obtained results consistent with our experiments. While we are able to give a good interpretation of our results within the framework of FRG, we have softened any claims on the interpretation of the actual microscopic mechanism in the actual material.

Reviewer #3 (Remarks to the Author):

We sincerely thank the referee for the careful reading of our manuscript and for the constructive comments that have helped us improve the clarity and precision of our work. We appreciate the reviewer's detailed comparison with previous studies on Ti-based kagome compounds and their insightful suggestions regarding both experimental methodology and data interpretation. We have carefully revised the manuscript and provide below a point-by-point response to each of the raised concerns.

1. The use of polarization-dependent ARPES to elucidate orbital contributions and autocorrelation methods to demonstrate electronic nematicity has been previously established in related kagome metals. Notably, orbital-selective behavior was reported in $RbTi_3Bi_5$ (Nat. Phys. 19, 1827, 2023), and clear autocorrelation evidence of nematicity was provided in the same compound (Nat. Commun. 14, 4892, 2023). The current manuscript does not appear to significantly advance beyond these earlier works. Especially, the autocorrelation ARPES result (Nat. Commun. 14, 4892 (2023)) is better than that in this manuscript.

We thank the referee for this valuable remark. We fully acknowledge the important prior work on $RbTi_3Bi_5$, which indeed established orbital-selective behaviour and electronic nematicity through polarization-dependent ARPES and autocorrelation analysis. The key advance of our present study lies in the *robustness* and *temperature stability* of the nematic state in $CsTi_3Bi_5$. While in $RbTi_3Bi_5$ the C_2 anisotropy is known to vanish around 200 K [Phys. Rev. B **112**, L041122], in our Cs-based compound the twofold-symmetric features persist at least up to room temperature. This persistence indicates a markedly more robust nematic response, suggesting that nematicity in $CsTi_3Bi_5$ is not a marginal or low energy fluctuation-driven effect, but rather an intrinsic and stable property of the electronic structure. The higher temperature range over which nematicity survives has important implications for understanding the microscopic mechanism as detailed in our many-body analysis at intermediate coupling which predicts PI with transition temperatures above RT. It points to an enhanced coupling between orbital and lattice degrees of freedom in $CsTi_3Bi_5$, possibly that may stabilise nematic order against thermal fluctuations. This robustness discriminates $CsTi_3Bi_5$ from its sister compound $RbTi_3Bi_5$, where comparable experimental studies have already been available (see reply to concern No. 4) and renders it a unique platform among Ti-based kagome metals for investigating the interplay between nematicity and topology in a regime accessible to practical conditions.

We thank the referee for raising this important point. In the discussion section of the paper, we have highlighted this remarkable difference between Ti-based kagome metals.

2. In the response to the Reviewer#1, the authors claim that “Ti-based kagome samples are highly reactive, and even mild thermal treatment in an inert Argon atmosphere can cause sample degradation”. However, previous work on RbTi₃Bi₅ (Supplementary Fig. 14 in Nat. Commun. 14, 4892 (2023)) showed clear C2 anisotropy persists at 200 K with negligible aging effect. This discrepancy requires clarification. Additionally, as shown in Extended Data Fig. 7, the autocorrelation maps at 16 K, 250 K and RT are much clearer than these at 90 K, 123 K, 200 K and 270 K. This non-monotonic changes in autocorrelation clarity across temperatures are not convincingly explained by aging effect alone.

Our statement regarding sample degradation reflects our specific experimental experience with CsTi₃Bi₅, rather than a generalised conclusion about all Ti-based kagome compounds. Indeed, we observed that CsTi₃Bi₅ surfaces are particularly sensitive to residual gases and mild thermal exposure, which can rapidly alter the surface spectral weight distribution.

However, we make the referee notice that we have clarified this point in the last revised manuscript and explicitly shown the aging effects in the rebuttal letter of our last time. The non-monotonic changes are also related to the causality of the cleave, as it is perfectly natural and expected for exfoliable crystals. Therefore, if a cleave is not ideal and perfectly flat, this is naturally causing electron-impurity scattering which, as a consequence of a contribution in the imaginary part of the self energy, causes a broadening of the signal (note that the signal is still good in the ARPES so it is a priori extremely difficult to understand which cleave is better).

We thank the referee for pointing out this missing piece of information. We have added an additional explanation to the caption of Extended Data Fig. 7 (now Supplementary Figure 13) for clarification.

3. The anisotropy observed in autocorrelation maps could arise from matrix element effects rather than intrinsic electronic ordering. To exclude this possibility, the authors should perform measurements with rotated sample orientations to verify whether the scattering vectors rotate accordingly, which is a standard validation step currently missing in the manuscript.

The measurements with rotated sample orientations have indeed been performed and were included in the Extended Figure 5 of the original submission, version 1 and they have not changed since then. As shown there, upon rotating the sample, the features in the autocorrelation maps (ACMs) rotate correspondingly, demonstrating that the observed anisotropy follows the crystallographic axes and is therefore intrinsic rather than arising from matrix element effects. There is also another methodology to do this, by using the sum of both circular polarizations to make the light as unpolarised as possible, and we did this too (and also this has been done for both directions).

Additionally, we have analyzed the peak positions of the ARPES intensities at the Fermi level for nominally equivalent high symmetry directions in Extended data Fig. 4 revealing a symmetry breaking shift of the Fermi contours along the two lines. Since this signature is independent of the absolute intensities and hence independent of matrix element effects (unlike the ACMs), we are convinced that symmetry breaking in the ACMs is stemming from a nematic state of the sample.

4. In their response to the Review#2, the authors state that “the only ARPES data available to date [Nat. Phys. 19, 1827 (2023)] for RbTi₃Bi₅ not for CsTi₃Bi₅, Nat

Commun. 14, 4089 (2023)] does not report a nematic state and focus only on the topological features of the normal state band structure”. This claim is inaccurate. In fact, the ARPES study published in Nat. Commun. 14, 4089 (2023) provides clear evidence of electronic nematicity via autocorrelation analysis (see Figures 5 and S11–S16), with data quality and resolution that appear superior to those presented in this work. Moreover, the authors emphasize that “As a matter of fact our study is the first to report nematicity in CsTi₃Bi₅ Ti-based kagome material from ARPES”. Given the strong electronic similarities between CsTi₃Bi₅ and RbTi₃Bi₅, and considering that robust nematicity has already been established in the latter through ARPES measurements, insisting on being “first” for the Cs compound seems unnecessary and does not substantively strengthen the manuscript’s contribution.

Our intention was not to overlook this important result, but to emphasise that those data concern the *Rb*-based compound. Nat Commun. 14, 4089 (2023) is indeed the only ARPES study of *CsTi₃Bi₅* and does not report nematic symmetry breaking in the spectra or ACMs. But the referee is absolutely correct that Nat. Commun. 14, 4892 (2023) provides this missing data for the *Rb*-based samples. We apologize for not making this utterly clear.

However, the differences between the two compounds are not minor: they significantly affect the Fermi surface topology [arXiv:2401.13628, Chinese Physics B33(10) (2024)] and the robustness of nematicity, which in *CsTi₃Bi₅* persists up to room temperature while diminishing at around 200K in *RbTi₃Bi₅* [Phys. Rev. B **112**, L041122]. Notably, theoretical calculations find a strong similarity between the two compounds (as mentioned by the referee), which requires a careful examination of the validity of the employed theoretical model by means of experimental probes to make sophisticated claims on the microscopic order parameter. This was carried out in this manuscript by a combined polarized ARPES and DFT+FRG approach.

We have therefore adjusted the phrasing to avoid unnecessary emphasis on priority, while clearly delineating the novelty of our findings within the broader Ti-based kagome family and discriminating our work from earlier works on *Rb*-samples.